# Significant contribution of subseafloor microparticles to the global manganese budget

Go-Ichiro Uramoto [1,2], Yuki Morono [1,3], Naotaka Tomioka [1], Shigeyuki Wakaki[1], Ryoichi Nakada [1], Rota Wagai[4], Kentaro Uesugi [5], Akihisa Takeuchi[5], Masato Hoshino[5], Yoshio Suzuki[5,6], Fumito Shiraishi [7], Satoshi Mitsunobu [8], Hiroki Suga[7,12], Yasuo Takeichi [9], Yoshio Takahashi[10] & Fumio Inagaki [1,3,11]

Ferromanganese minerals are widely distributed in subseafloor sediments and on the seafloor in oceanic abyssal plains. Assessing their input, formation and preservation is important for understanding the global marine manganese cycle and associated trace elements. However, the extent of ferromanganese minerals buried in subseafloor sediments remains unclear. Here we show that abundant ($10^8$–$10^9$ particles cm$^{-3}$) micrometer-scale ferromanganese mineral particles (Mn-microparticles) are found in the oxic pelagic clays of the South Pacific Gyre (SPG) from the seafloor to the ~100 million-year-old sediments above the basement. Three-dimensional micro-texture, and major and trace element compositional analyses revealed that these Mn-microparticles consist of poorly crystalline ferromanganese oxides precipitating from bottom water. Based on our findings, we extrapolate that 1.5–8.8 × $10^{28}$ Mn-microparticles, accounting for 1.28–7.62 Tt of manganese, are globally present in oxic subseafloor sediments. This estimate is at least two orders of magnitude larger than the manganese budget for nodules and crusts on the seafloor. Subseafloor Mn-microparticles thus contribute significantly to the global manganese budget.

[1] Kochi Institute for Core Sample Research, Japan Agency for Marine-Earth Science and Technology (JAMSTEC), Kochi 783-8502, Japan. [2] Center for Advanced Marine Core Research, Kochi University, Kochi 783-8502, Japan. [3] Research and Development Center for Submarine Resources, JAMSTEC, Yokosuka 237-0061, Japan. [4] Institute for Agro-Environmental Sciences, National Agriculture and Food Research Organization, Tsukuba 305-8604, Japan. [5] Japan Synchrotron Radiation Research Institute, Hyogo 679-5198, Japan. [6] Graduate School of Frontier Sciences, The University of Tokyo, Kashiwa 277-8561, Japan. [7] Department of Earth and Planetary Systems Science, Hiroshima University, Hiroshima 739-8526, Japan. [8] Department of Environmental Conservation, Graduate School of Agriculture, Ehime University, Matsuyama 790-8566, Japan. [9] Institute of Materials Structure Science, High Energy Accelerator Research Organization, Tsukuba 305-0801, Japan. [10] Department of Earth and Planetary Science, The University of Tokyo, Tokyo 113-0033, Japan. [11] Research and Development Center for Ocean Drilling Science, JAMSTEC, Yokohama 236-0001, Japan. [12] Present address: Department of Earth and Planetary Science, The University of Tokyo, Tokyo 113-0033, Japan. These authors contributed equally: Go-Ichiro Uramoto, Yuki Morono. Correspondence and requests for materials should be addressed to F.I. (email: inagaki@jamstec.go.jp)

Ferromanganese mineral deposits are abundant in marine and terrestrial environments; thus, understanding the redox-sensitive dynamics and budget of ferromanganese minerals are important for understanding the global cycling of manganese and numerous associated trace elements[1–6]. Since the discovery of manganese nodules during the 1872–76 voyage of HMS *Challenger*[7], ferromanganese mineral deposits have been widely observed in marine sediments on the seafloor[1–3]. The most extensive manganese minerals deposits occur on abyssal plains, including those under open-ocean gyres[2], where both organic matter photosynthetic primary production and sedimentation rates are particularly low[8,9]. In general, manganese nodules grow slowly on the oxic seafloor over geologic time, accumulating concentric deposits consisting primarily of manganese and iron accompanied by various trace metals, including nickel, cobalt, lithium, molybdenum, zirconium, and rare-earth elements (REEs) [1–3]. Consequently, these mineral deposits are commonly found as spherical concretions of centimeter-to-millimeter size, which often form pavements over wide areas of the ocean floor[1–3,10–12]. In addition, ferromanganese mineral deposits have been observed in the underlying subseafloor sediment in the form of manganese nodules, including sub-millimeter-size micronodules[13–15].

In 2010, Integrated Ocean Drilling Program (IODP) Expedition 329 drilled the seafloor in the ultra-oligotrophic region of the South Pacific Gyre (SPG) to investigate the deep-sea sedimentary environment (Supplementary Figure 1 and Supplementary Table 1)[8,16]. Geochemical and microbiological studies of sediment samples cored from six SPG drill sites revealed the presence of dissolved oxygen and aerobic microbial communities throughout the sedimentary sequence, from the present-day seafloor to the mid-Cretaceous sediment layer above the crustal basement (Supplementary Figure 2). These organic-poor oxic sediments represent up to ~44% of the Pacific and ~37% of the global ocean (Supplementary Figure 1)[8]. The SPG sediments were found to consist mainly of zeolitic metalliferous pelagic clay, and massive manganese nodules were observed on the seafloor

and locally within sediments at these drilling sites (Supplementary Figure 2)[16].

In this work, to better characterize micrometer-scale sediment microstructures in the entirely oxic SPG core samples, we analyzed and compared marine sediment core samples in various redox condition through continental margin to open ocean gyre (Supplementary Table 2) using a resin-embedding technique[17] involving a biological method that retains the fine texture of both mineral grains and organic materials in sediments at the submicrometer-scale. We demonstrate the presence of remarkable numbers of micrometer-scale ferromanganese mineral particles (Mn-microparticles) in the deep-sea oxic pelagic clay of the SPG. We also develop a technique that combines sequential density concentration[18] and flow cytometry/particle sorting[19] to separate Mn-microparticles from the sediment matrix, which enables further analysis of the texture and composition of these microparticles. We then determine the manganese content of the Mn-microparticles, estimating that Mn-microparticles harbor notably higher amounts of manganese in subseafloor environments than indicated in previous manganese budget estimates based on seafloor nodules and crusts data[20]. Finally, we examine the formation and preservation of micrometer-scale ferromanganese minerals in an oxic deep subseafloor environment in order to improve our understanding of the global manganese budget.

## Results

**Microstructure and mineral characteristics of microparticles.** Scanning electron microscope (SEM) imaging results for resin-embedded samples from various marine environments showed that submicrometer-scale to micrometer-scale mineral particles (Figs. 1 and 2, Supplementary Figures 3 and 4) (0.1–40 µm, average diameter 3.4–4.4 µm, Supplementary Figure 5) were abundant in the sediments. These particles were broadly classified into two types: (1) a morphology resembling a clump of tangled

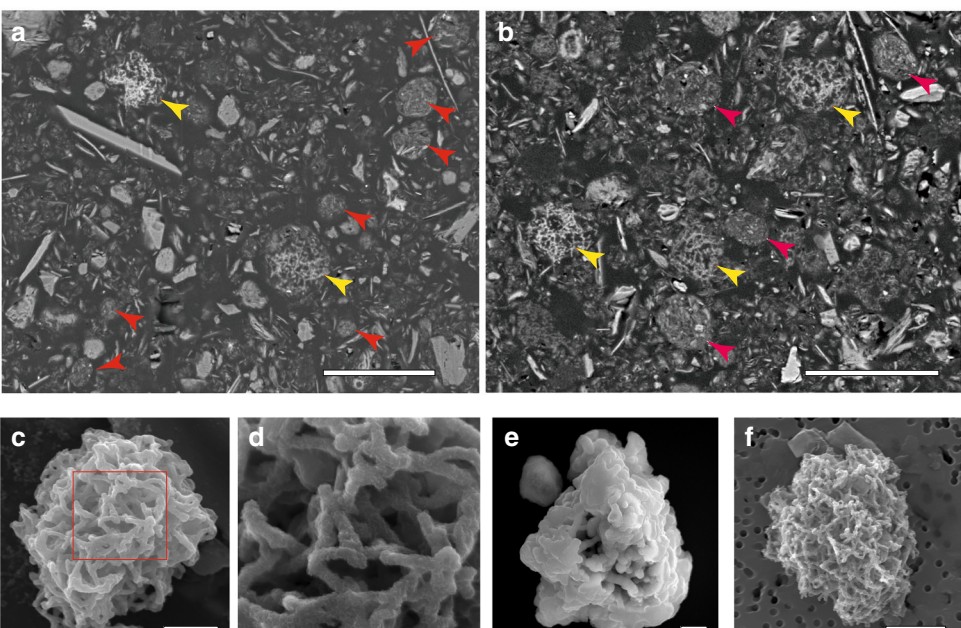

**Fig. 1** Representative electron micrographs of microparticles in sediment samples. **a, b** Cross-sectional scanning electron microscopy (SEM) images of resin-embedded oxic pelagic clay. Arrows indicate Mn-microparticles (yellow) and clay-microparticles (red) (samples U1365C-1H-2 0/20 and U1367D-1H-2 20/40, respectively). Scale bars, 10 µm. **c** SEM image of a Mn-microparticle in a density-separated sample (sample U1365C-1H-2 0/20). Scale bar, 5 µm. **d** Enlargement of the area within the red square in (**c**) showing the tangled fibrous strands in a microparticle. Scale bar, 500 nm. **e, f** SEM images of Mn-microparticles in density-separated samples (samples U1365C-9H-3 35/55 and U1366F-1H-2 40/60, respectively). Scale bars, 5 µm

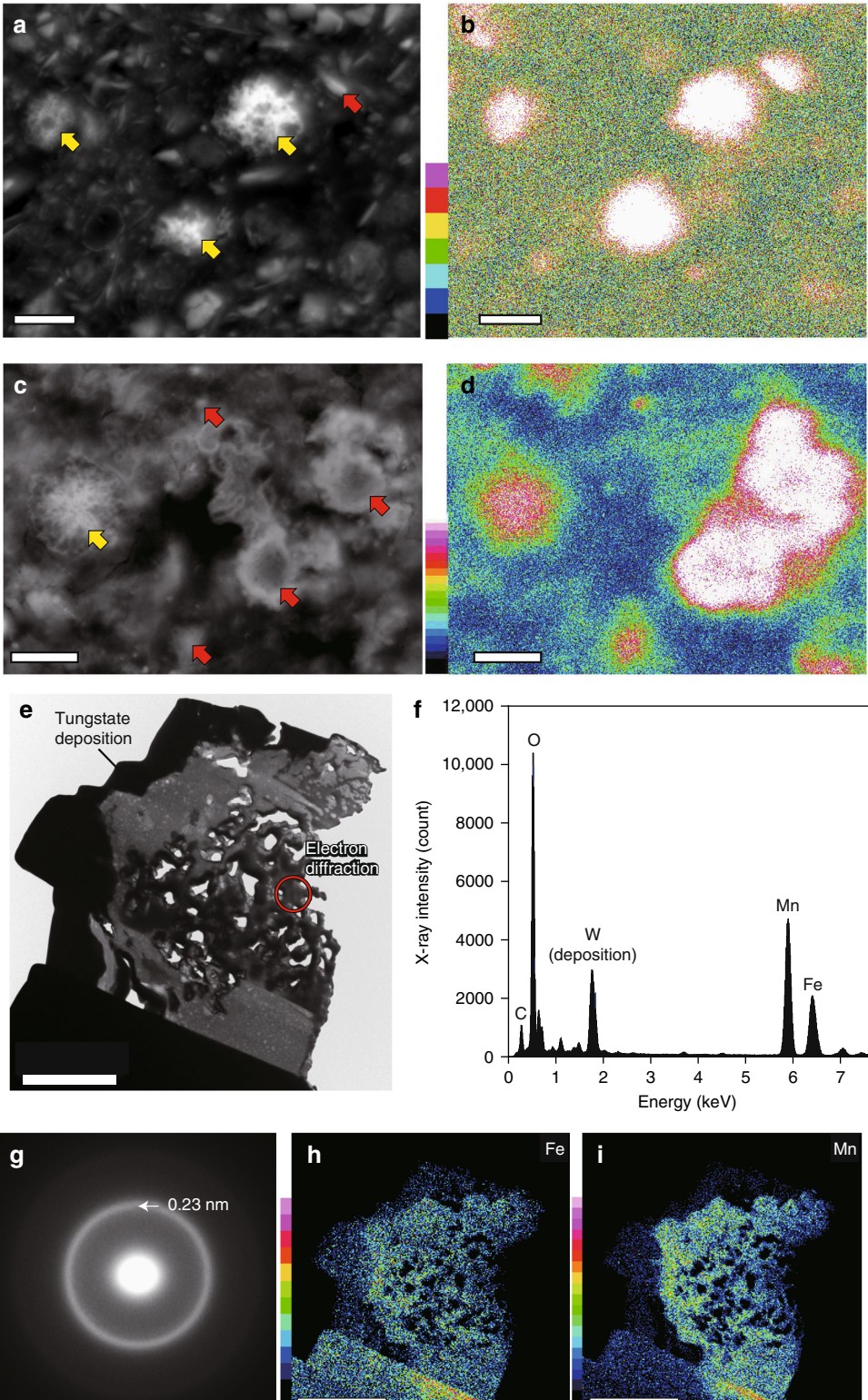

**Fig. 2** Composition and mineral characteristics of Mn-microparticles. **a–d** Representative elemental map of manganese in a resin-embedded sediment sample. Scale bars, 5 µm: **a** A back-scattered electron (BSE) image acquired by scanning electron microscopy (SEM) and **b** an energy-dispersive X-ray spectroscopy (EDS) elemental map of manganese for sample U1365D-1H-2 0/20; **c** A back-scattered electron (BSE) image acquired by SEM and **d** an elemental map of manganese for sample U1365D-9H-3 35/55; yellow arrows indicate Mn-microparticles, and red arrows indicate manganese-concentrated particles without concentric growth structure and fibrous structure in the samples herein. **e–i** Representative transmission electron microscopy (TEM) analysis results for Mn-microparticles. Scale bars, 2 µm: **e** cross-sectional TEM image of an FIB-cut Mn-microparticle, and **f** an EDS spectrum; the W peak is an artifact derived from contaminated W-deposition during focused ion-beam (FIB) fabrication. **g** Electron diffraction pattern. **h**, **i** Elemental maps for Fe and Mn, respectively, in sample U1365C-1H-2 0/20. Based on the EDS spectrum and electron diffraction pattern, the main constituent is poorly crystalline ferromanganese oxide

fibrous strands of manganese and iron minerals (~0.2 µm in diameter and ~1 µm long; Mn-microparticles; Figs. 1 and 2, Supplementary Figure 3, Supplementary Movies 1 and 2), and (2) clumps of clay-size platelet silicate minerals (<0.1 µm in thickness and ~1 µm in diameter) (clay-microparticles; Fig. 1a and b and Supplementary Figures 3 and 4). Elemental maps of manganese in the resin-embedded samples generated using SEM energy-dispersive X-ray spectroscopy (EDS) showed high contents of manganese in these microparticles (Fig. 2a–d). Transmission electron microscope (TEM) and synchrotron-based three-dimensional (3D) X-ray-computed microtomography imaging showed that these Mn-microparticles lacked cores and concentric laminations (Fig. 2c; Supplementary Movies 1 and 2). In TEM analysis (Fig. 2e–i), the electron diffraction patterns (ED) of the Mn-microparticles showed broad diffraction rings indicating the existence of a poorly crystalline manganese oxide (vernadite). TEM-EDS analysis revealed that the Mn-microparticles also contained large amounts of iron. The iron was incorporated in manganese oxides, according to X-ray elemental maps. TEM-EDS-ED analysis showed that the clay-microparticles consisted mainly of a nontronite-like iron-bearing clay mineral, with limited "hot spots" of higher iron and manganese concentrations (Supplementary Figure 4).

**Abundance of microparticles in marine sediments**. The abundance and distribution of these microparticles were assessed via SEM-image analysis (Supplementary Figure 6). In the sediment column, clay-microparticles had abundances of $10^6$–$10^9$ cm$^{-3}$ (average $4.99 \times 10^8$ cm$^{-3}$) in both continental margin and pelagic sediment samples. Mn-microparticles occurred at $10^8$–$10^9$ cm$^{-3}$ (average $3.34 \times 10^8$ cm$^{-3}$) only in the SPG oxic pelagic clay (Fig. 3). Neither type of microparticle was observed in the SPG calcareous ooze (Fig. 3).

**Elemental composition of Mn-microparticle**. To determine the origin and/or diagenesis of Mn-microparticles in sediments, we analyzed the elemental compositions of the Mn-microparticles by separating them from the surrounding sedimentary matrix using combined sequential density concentration[18] and flow

cytometry/particle sorting techniques[19] (Supplementary Figure 7). Inductively coupled plasma-mass spectrometry (ICP-MS) analysis of the separated Mn-microparticles revealed various metallic elements, including rare metals and REEs (Supplementary Table 3). The Mn-microparticles contained average iron and manganese mass contents of 18 and 17.9 pg/particle, respectively, five to ten times the mass of the other major elements (Fig. 4a and b, Supplementary Figure 8). The Mn-microparticles contained particularly high amounts of manganese, which accounted for an average of 42% of the total manganese in the bulk SPG sediment samples[21] (Fig. 4a–d). The manganese percentage in the Mn-microparticles was high (>40–70%) at shallow depths, but manganese in the Mn-microparticles comprised <1.8% of the total manganese mass in bulk sediment samples above the basement (Fig. 4d). We also found manganese-concentrated particles without concentric growth structure nor fibrous structure (structureless manganese particles) which were outside of Mn-microparticles with tangled fibrous strands (Fig. 2a–d; Supplementary Figure 9). These structureless manganese particles were submicrometer to micrometer in size in shallow sediments (Fig. 2a–d; Supplementary Figure 9a–d) and up to several tens of micrometers in sediments above the basement (Supplementary Figure 2c and d). In oxic sediments, both Mn-microparticles and structureless manganese particles constituted significant fractions of the manganese mineral content. However, there was considerable size variation in the structureless manganese particles, which prevented precise separation and purification by optical characteristics and mineral particle sizes using the flow cytometry-based particle sorting technique employed in this study. This, in turn, prevented a more in-depth analysis of the structureless manganese particles (Supplementary Figure 5). Therefore, we focused on the recoverable fibrous Mn-microparticles in this study.

As shown in the ternary diagram of $(Co + Cu + Ni) \times 10$–Fe–Mn[22] from Mn-microparticles in Fig. 4e, the composition of these elements was closely related to the ferromanganese minerals of hydrogenetic origin; the oldest sediment sample above the basement (Sample U1365C-9H-3 35/55) with high iron content varied between hydrogenetic and hydrothermal origin (Fig. 4e). The Mn-microparticle REE concentration

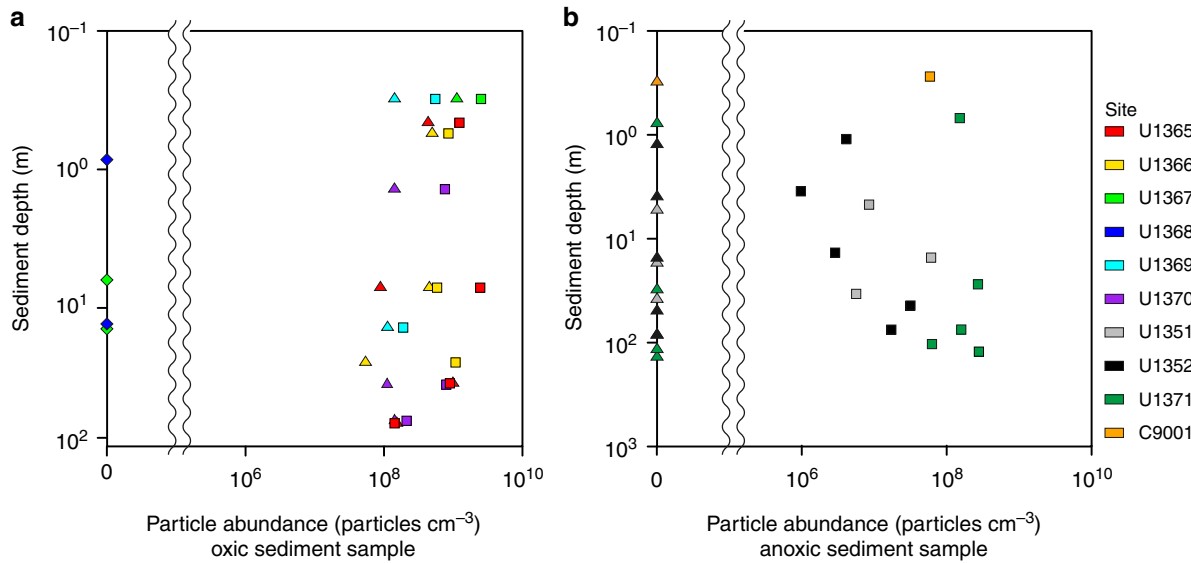

**Fig. 3** Number of microparticles in sediment samples. **a** Depth profiles of numbers of Mn-microparticles in pelagic oxic sediment samples from the South Pacific Gyre. **b** Depth profiles of numbers of clay microparticles in anoxic sediment samples from the South Pacific and continental margin. In **a** and **b**, colored triangles indicate Mn-microparticles, colored squares denote clay-microparticles, and colored diamonds indicate calcareous ooze samples without microparticles

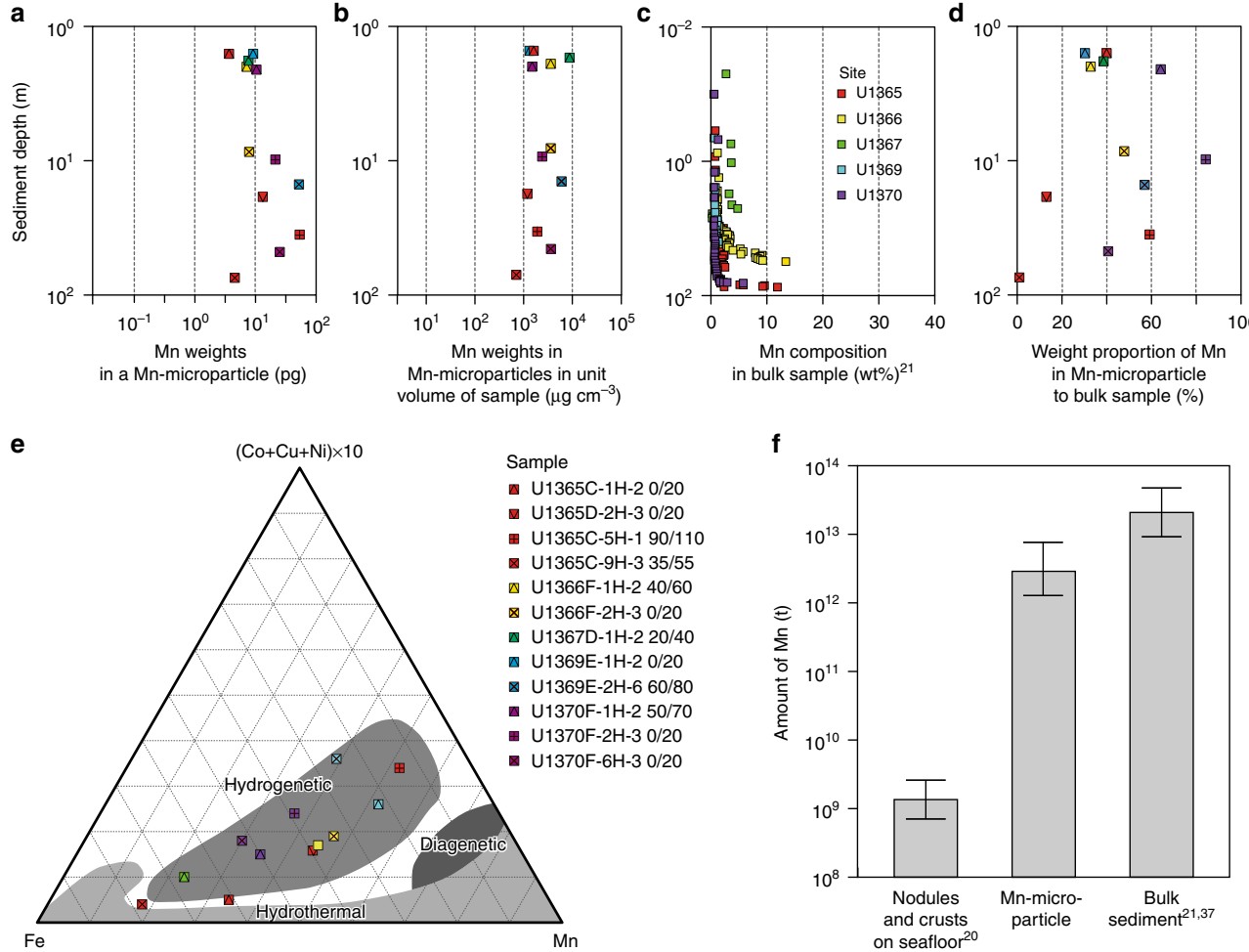

**Fig. 4** Origins and budget of manganese in Mn-microparticles in oxic pelagic sediments. **a–d** Depth profiles of mass and proportion of manganese in Mn-microparticles in sediment samples: **a** average mass of major elements in a Mn-microparticle; **b** average mass of manganese in Mn-microparticles in a unit volume of sample; **c** major-element composition of the bulk sample[21]; and **d** proportion of Mn-microparticle manganese mass to bulk sample mass. **e** Fe–Mn–(Co + Cu + Ni) × 10 ternary diagram[22] for the Mn-microparticles. **f** Comparison of the manganese budgets estimated for Mn-nodules and crusts on the seafloor[20], Mn-microparticles in oxic pelagic clay, and a bulk sample of oxic pelagic clay[21]. Whiskers represent maximum and minimum estimates

patterns showed a decrease in the cerium anomaly (Ce/Ce*) with increasing depth (Supplementary Figure 10a and b).

**Redox state of bulk sediments and Mn-microparticles**. The X-ray absorption near-edge structure (XANES) spectra at the Mn K-edge (Fig. 5a) of the oxic bulk sediment samples herein exhibited absorption maxima similar to that of reference $MnO_2$ minerals, such as δ-$MnO_2$ and birnessite. Quantitative fitting of these spectra showed that Mn(IV) was the dominant manganese redox state in the oxic bulk sediments studied herein. The extended X-ray absorption fine structure (EXAFS) spectra of selected samples (Fig. 5b) also showed dominant Mn(IV) in the bulk sample studied herein. The XANES spectra at the Fe K-edge exhibited absorption maxima similar to that of reference illite and montmorillonite (smectite) (Fig. 5c).

Scanning transmission X-ray microscopy (STXM) analysis at the Mn L-edge and Fe L-edge was used to characterize the redox state of these elements in the Mn-microparticles (Supplementary Figure 11). The near edge X-ray absorption fine structure (NEXAFS) spectra of manganese featured peaks or shoulders at 639.5, 640.6, 641.4, and 643.1 eV and absorption patterns similar to spectra of Mn(II) reference materials (i.e., $MnCO_3$). However, repeated STXM measurement of Mn(IV)

reference minerals (i.e., δ-$MnO_2$ and birnessite) showed gradual changes in the manganese NEXAFS spectra (Supplementary Figure 11). The iron NEXAFS spectra contained peaks at 707.5 and 709.1 eV; the relative intensities of these peaks were similar to those in the spectra of a Fe(III) reference mineral (i.e., ferrihydrite).

**Thermodynamic properties of sediment porewater**. To explore the degree of manganese mineral saturation in porewater from the SPG oxic subseafloor environment, thermodynamic calculations were conducted for Mn(IV) and Mn(II) minerals based on the porewater chemical composition[16]. The resulting saturation index values ranged from 8.37 to 9.16 for the Mn(IV) minerals and from −0.54 to −1.38 for the Mn(II) minerals.

**Discussion**
Abundant Mn-microparticles have not been previously observed in oxic pelagic clay, nor are they expected. Fibrous ferromanganese oxide minerals, which are similar in morphology to the Mn-microparticle component minerals, have been reported on rock slabs exposed on the deep-sea floor at experimental scales; however, such minerals have not been observed within rock samples[23]. This study finds, importantly, that manganese is highly concentrated in Mn-microparticles within oxic sediments

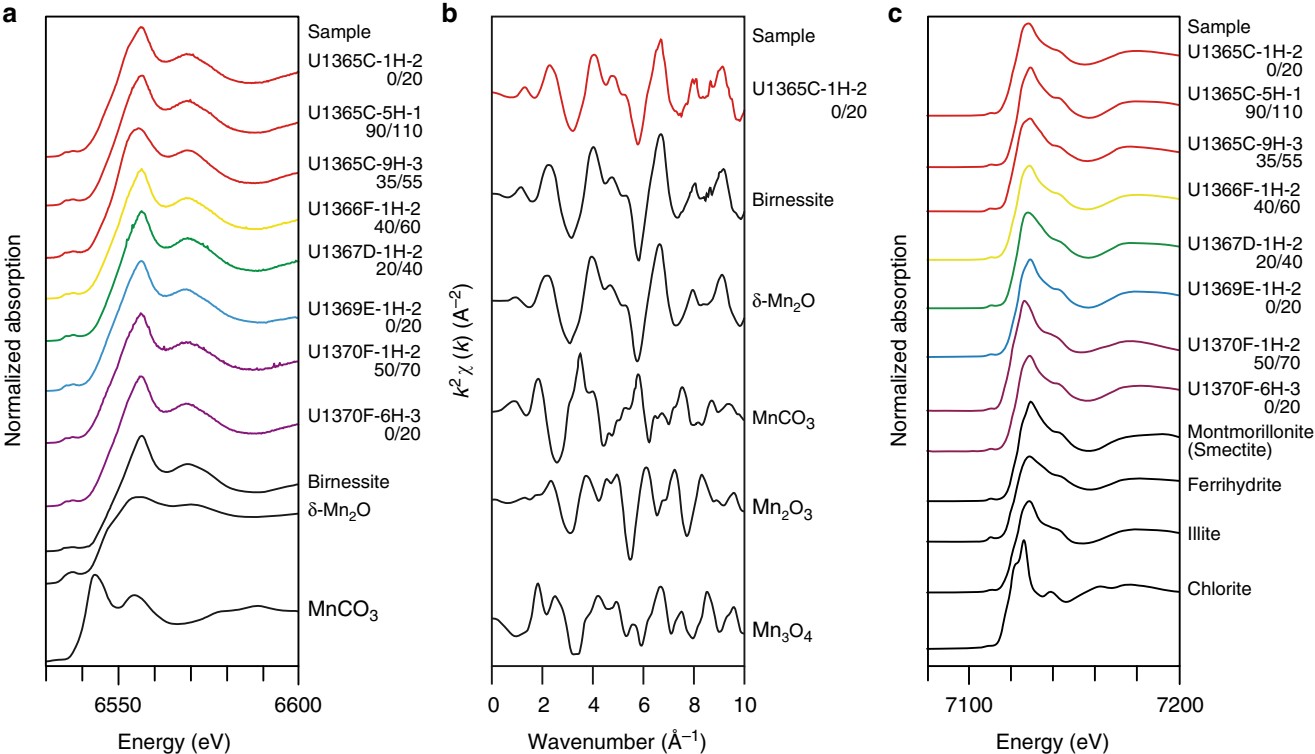

**Fig. 5** Spectroscopic characteristics of bulk samples in this study. **a**, **b** X-ray absorption fine structure (XAFS) results for selected bulk samples and standard samples at the Mn K-edge: **a** X-ray absorption near edge structure (XANES) spectra of selected and standard samples, **b** representative normalized $k^2$-weighted extended XAFS spectra for sample U1365C-1H-2 0/20 and standard samples. **c** XANES spectra of selected and standard samples at the Fe K-edge

(Fig. 2a and b), which constitutes a significant portion of the oceanic subseafloor sediment[8], and that these micrometer-scale mineral particles contribute significantly to the manganese budget on Earth.

We propose that Mn-microparticles in oxic SPG sediments are derived primarily from the hydrogenetic precipitation of manganese oxides from bottom water by using manganese derived from widespread transportation of eolian dust in the South Pacific Ocean[21] (Supplementary Figure 12). The (Co + Cu + Ni) × 10–Fe–Mn (Fig. 4e)[22] ternary diagram shows that Mn-microparticles are similar in composition to hydrogenetically precipitated ferromanganese crusts[6,24–28] and manganese nodules[26,29] on the seafloor. Moreover, REE concentration patterns in these Mn-microparticles reveal predominantly positive cerium anomalies (Supplementary Figure 10). Dissolved cerium in bottom seawater is easily oxidized and adsorbed in ferromanganese oxides[6] and/or is hydrolyzed and irreversibly scavenged from seawater on the surface of ferromanganese oxides[6]. The samples herein that feature positive Ce anomalies were formed through hydrogenetic precipitation in bottom water. However, it should be noted that hydrogenetic Mn-microparticles show varying metallic element composition (Fig. 4e). This could be due to remobilization of metals in association with the degradation of organic matter in subseafloor environments. This indication of metallic element mobilization in Mn-microparticles should be verified in the future through the sampling and analysis of sedimentary Mn-microparticles between continental margin and open ocean gyre. On the other hand, composition of the Mn-microparticles found in the oldest sediment (sample U1365C-9H-3 35/55) examined in this study show a close relationship with hydrothermal deposits[26,30–32] in the (Co + Cu + Ni) × 10–Fe–Mn diagram (Fig. 4e)[22],

indicating the involvement of hydrothermal processes in their formation. This hydrothermal effect can also be inferred from the negative Ce anomalies in the sample (Supplementary Figure 8b). Because of the slow cerium oxidation kinetics[33] and the rapid formation of ferromanganese minerals by hydrothermal processes[34], hydrothermal ferromanganese oxides deposited near oceanic ridges (e.g., East Pacific Rise) generally feature negative Ce anomalies[35–37]. Taken together, the hydrothermal influence on Mn-microparticles seems to change with the tectonic migration of the study sites from the oceanic ridge[8] (Supplementary Figure 12). Both the SPG sediments accumulated near the oceanic ridge and the Mn-microparticles formed in the ridge environment should be affected by hydrothermal plumes[38–41]. As the oceanic plate migrates from the oceanic ridge, the hydrothermal effect should diminish and hydrogenetic formation of Mn-microparticles in bottom water become dominant.

In this study, Mn-microparticles are not observed in sediments from continental margin or in oxic SPG calcareous oozes (Fig. 3). The formation process itself does not preclude precipitation in the oxic bottom water at continental margin. However, as the redox state in subseafloor environments at continental margin is reduced, ferromanganese oxides are presumed to be dissolved into porewater or precipitated as Mn(II) mineral (i.e., rhodochrosite)[42–44]. Regarding oxic calcareous oozes, there are two possible mechanisms that explain the absence of Mn-microparticles. In general, the sedimentation rate above the calcium carbonate compensation depth (CCD) is higher than that below the CCD[3]. In sediments above the CCD, the relative abundance of Mn-microparticles in sediments can be lower than the SEM-based detection limit for these mineral particles in this study. The other possibility involves the adsorption of manganese onto carbonate[45,46]. In this case, the formation of Mn-microparticles would be hampered by the

decreased concentration of dissolved manganese in the bottom water. Due to these two mechanisms, the existence of highly abundant Mn-microparticles is restricted at depths below the CCD in pelagic environments.

It should be noted that there is a potential mismatch in the spectroscopic analyses of manganese in bulk sediments and the separated Mn-microparticles (Fig. 5; Supplementary Figure 11). Bulk sample XAFS analysis indicates the manganese oxidized state, while separate Mn-microparticle STXM analysis indicates the manganese reduced state. As discussed above, the Mn-microparticles consist primarily of ferromanganese oxides. Thermodynamic calculations of the redox state of the SPG sediment porewater suggest that Mn(II) minerals are unsaturated, which precludes precipitation of Mn(II) minerals in the oxic subseafloor environment. In addition, repeated measurements of Mn(IV) reference minerals ($\delta$-$MnO_2$ and birnessite) show that the manganese redox state changes during STXM measurements (Supplementary Figure 11). Therefore, we conclude that (1) the primary redox state of manganese in the Mn-microparticles is oxidized and (2) photoreduction of manganese in Mn-microparticles may have occurred via irradiation by the intense X-rays during STXM measurements. The estimated photon flux in our spectroscopic measurements was $\sim 1 \times 10^{15}$ photons $s^{-1}$ $mm^{-2}$ in STXM[47] and $\sim 5 \times 10^{10}$ photons $s^{-1}$ $mm^{-2}$ in XAFS[48]. In contrast, iron was consistently in an oxidized state in the STXM analyses. Considering the redox-sensitive nature of manganese[1–3], we thus conclude that the photoreduction of manganese occurred during STXM measurements due to intense X-ray irradiation. This indicates that careful adjustment of the photon flux and X-ray exposure time is necessary to ensure precise spectroscopic determination of the manganese redox state in Mn-microparticles.

Our data also suggest that Mn-microparticles are not formed within the sediment by diagenesis, based on the (Co + Cu + Ni) × 10–Fe–Mn diagram[22] (Fig. 4e). The relative consistency in the Mn-microparticle average size throughout this study (Supplementary Figure 5) indicates that the Mn-microparticles have been stable since their deposition over ~100 Ma. This stability can be used to infer the flux of Mn-microparticles into the oxic sediments. According to the Mn-microparticle abundance in the sediments (Fig. 3) and sedimentation rate at the corresponding site (Supplementary Figure 2), the calculated flux of the Mn-microparticles into the sediment is $108 \pm 35$ microparticles $cm^{-2}$ $day^{-1}$.

Lastly, this study has important implications for understanding the global distribution and budget of manganese contained in Mn-microparticles within oxic subseafloor environments (Fig. 4f). If we assume that the number of Mn-microparticles in the SPG pelagic clay is roughly constant at $3.34 \times 10^8$ $cm^{-3}$, and account for gradual decreases in manganese content in Mn-microparticles with sediment depth (>40–70% in shallow sediments and <1.8% in deep sediments) (Fig. 4d) and the global distribution of oxic pelagic clay[8,49] (Supplementary Figure 1), an estimated $1.5$–$8.8 \times 10^{28}$ Mn-microparticles are present in the oxic pelagic clays of the world's oceans, corresponding to 1.28–7.62 Tt of manganese (Fig. 4f). Given the manganese contained in pelagic sediments[21,38], we estimate that 9.2–47.4 Tt of manganese is present, suggesting that roughly 10–20% of manganese in oxic pelagic clays is in the form of Mn-microparticles. Although our estimates do not include manganese from outside of Mn-microparticles, such as structureless manganese particles, our manganese budget estimate from Mn-microparticles is at least 2–3 orders of magnitude higher than those presented in previous studies based on seafloor nodules and crusts, which propose budgets of 0.706–2.60 Gt of manganese[20] (Fig. 4f). Along with the examination of manganese input, formation, and preservation,

the discovery herein of abundant Mn-microparticles provides new insight into the global manganese budget.

## Methods

**Sample description.** All pelagic sediment core samples from the SPG were collected during IODP Expedition 329 (Supplementary Figure 1 and Supplementary Table 1)[16]. The SPG samples from Sites U1365–U1367 and U1369–U1370 consisted principally of zeolitic and/or metalliferous (iron-manganese) pelagic clays, whereas those from Sites U1367–U1368 contained calcareous nannofossil oozes. The sediment core samples from Site U1371 consisted of clay-bearing diatom ooze and pelagic clay (Supplementary Figure 2). Pebble-sized manganese nodules were observed on the seafloor and locally in sediments at Sites U1365–U1367 and U1369–U1370, but were absent at Site U1368, at which the water depth was shallower than the expected CCD, and at Site U1371, which lay outside of the SPG[16]. The water depths at these sites ranged from 4289 m (U1367) to 5695 m (U1365) below sea level. The basement ages ranged from the late Eocene (~33.5 Ma, U1367) to the mid-Cretaceous (~100 Ma, U1365)[16]. The average sedimentation rates calculated from the total sediment thicknesses and age models[16,50] at each SPG site were extremely low, ranging from 0.3 to 1 m $Myr^{-1}$. At Site U1371[51,52], the average sedimentation rate for the pelagic clay interval (0.33 m $Myr^{-1}$) was similar with those at other SPG sites, but that of the diatomaceous clay interval was notably higher (10 m $Myr^{-1}$). At SPG sites, dissolved oxygen was detected throughout the sediment column, from the seafloor to the sedimentary basement, suggesting an extremely low rate of oxygen consumption by subseafloor microbial communities[8] (Supplementary Figure 2). In contrast, at Site U1371, dissolved oxygen decreased to below the detection limit within 0.9 m below the seafloor[8].

The continental margin sediment samples used in this study were collected during IODP Expedition 317 at Sites U1351 and U1352 in the Canterbury Basin off New Zealand[53] and at Site C9001 (i.e., IODP Site C0020) during JAMSTEC cruise KY11-E06 off of the Shimokita Peninsula, Japan[54]. The samples consisted of sandy silt, silt, sandy to silty clay, and diatomaceous clay. Siliciclastic sediments from the Canterbury Basin consisted dominantly of quartz and feldspar, with common-to-rare mica and clay minerals. The diatomaceous sediments collected from Site C9001 consisted mainly of clay and diatoms. The authigenic minerals in the continental margin sediments were primarily opaque minerals (i.e., framboidal pyrite). The average sedimentation rates, which were calculated from total sediment thicknesses and age models determined on the basis of microfossil assemblages and ash layers, ranged from 200 to 620 m $Myr^{-1}$.

**Resin-embedding processing.** A total of 31 sediment samples (Supplementary Table 2) were selected for the resin-embedding procedure[17]. Sediments were subsampled using a plastic straw (3.8 mm in diameter) from the bulk sample. Each sub-sample was cut into lengths of 1–2 mm and placed in plastic mini-containers, which were immersed in 2% agarose at 37 °C, and then cooled to 4 °C to solidify the agarose. The agarose-embedded sample plugs were gradually dehydrated by sequential soaking in a series of ethanol solutions as follows: 30%, 50%, 70%, and 80% ethanol for 15–20 min at 4 °C, then 90% and 95% ethanol for 15–20 min at room temperature, and then three times with 100% ethanol for 15–20 min at room temperature. These steps were followed by three washes with n-butyl glycidyl ether (QY-1; Nissin EM, Tokyo, Japan) for 15–20 min at room temperature. The sediment plugs were then sequentially immersed in mixtures of QY-1 and Quetol 651 epoxy resin (TAAB, Aldermaston, UK) as follows: a 3:1 mixture for 1 h, a 2:1 mixture for 2 h, a 1:1 mixture for 3 h, a 1:2 mixture overnight, a 1:3 mixture for 8 h, and 100% resin overnight. Finally, the sediment plugs were heated in an oven at 37 °C for 12 h, followed by a gradual temperature increase to 60 °C for 24 h, after which the resin was cured in an oven at 60 °C for 24 h.

**Scanning electron microscopy.** Resin-embedded sediment samples were cut with a diamond knife installed on an RX-860 Rotary microtome (Yamato Kohki Industrial Co. Ltd., Japan). The microtome-cut sample surfaces were coated with carbon using a CADE-E carbon coater (Meiwafosis, Japan). Images of the coated samples were obtained with a JSM-6500F SEM (JEOL, Tokyo, Japan). EDS analysis was performed at 10 kV to measure the qualitative composition of the microparticles and elemental distribution in the samples. Backscattered electron images were obtained at 7 kV to measure the number and diameter of microparticles within a 250 × 400 μm image window. The number of microparticles in the SEM image data was converted into a volume concentration through image analysis (Supplementary Figure 4), which was based on the assumption that the 2D image-based areal ratio was equal to the volume ratio[55]. We then estimated the number of microparticles per unit volume. The global number of microparticles was estimated based on the global distribution of oxic sediments in pelagic environments (Supplementary Figure 1)[8] and the proportion of pelagic clay in sediment cores[49].

**Density concentration.** Twelve samples were selected for sequential density concentration (Supplementary Table 2) using sodium polytungstate (SPT; Sometu, Germany) solutions adjusted to densities of 2.20, 2.35, 2.61, 2.66, 2.70, and

$2.80\ g\ cm^{-3}$. First, 15 g sub-samples were mixed with distilled water in a centrifuge bottle and shaken at 500–1000 rpm for 5 min. Then, the samples were centrifuged at $2200g$ for 15 min, and the supernatant was removed. These steps (resuspension in distilled water, centrifugation, and supernatant removal) were repeated until the electric conductivity (EC) of the supernatant was below $70\ \mu S\ cm^{-1}$. Next, the remaining materials were suspended in a SPT solution adjusted to a density of $2.80\ g\ cm^{-3}$ and centrifuged at $2200g$ for 15 min, after which the floating materials were resuspended and centrifuged in sequential lower density SPT solutions. The remaining materials were resuspended in distilled water, centrifuged at $2200g$ for 15 min, and the supernatant removed. These steps were repeated to remove the SPT $(EC < 70\ \mu S\ cm^{-1})$. We conducted further density concentration with $3.0\ g\ cm^{-3}$ SPT solution to confirm that all materials floated from the samples. Subsequently, we filtered the remaining material from each step and confirmed via SEM that Mn-microparticles appeared when the SPT solution density exceeded $2.35\ g\ cm^{-3}$, and the proportion increased when the SPT solution density exceeded $2.7\ g\ cm^{-3}$. Therefore, the isolated samples were estimated to have a density range of $2.35$–$3.0\ g\ cm^{-3}$; in subsequent sample processing and analyses, we used the remaining materials processed with SPT solutions with densities exceeding $2.70\ g\ cm^{-3}$ SPT.

**Flow cytometry and particle sorting**. Density-concentrated samples were suspended in 0.1 M NaCl solution and further processed; Mn-microparticles were enriched using a Moflo XDP cell sorter (Beckman Coulter, Brea, CA, USA). During Mn-microparticle enrichment, forward-scatter and side-scatter signal from a 488-nm laser, as well as the absence of fluorescence under a 355 nm laser, were found to be effective for selectively harvesting up to 80–95% of sorted the Mn-microparticles. Then, $1.25 \times 10^{5}$ microparticles were sorted into a pre-washed ($HNO_3$ and HF) plastic vial containing 50 $\mu$L HEPES buffer (10 nM, pH 7.4).

**Synchrotron-based X-ray micro-computed tomography**. X-ray $\mu$CT imaging was performed using the Super Photon ring-8 (Spring-8) GeV at the synchrotron radiation facility in Hyogo, Japan[56]. For the imaging, a small amount of separated sample was mounted on gecko adhesive[57] (Nitto Denko Co. Ltd., Japan) and scanned at an X-ray energy of 8 keV with an exposure time of 150 ms and rotary table angular resolution of 0.2°. Scanned X-ray $\mu$CT data were reconstructed via the filter backprojection method to produce cross-sectional Tag Image File Format (TIFF) images at 70 nm pixel size[56]. 3D visualization of the $\mu$CT images was performed using OsiriX image processing software.

**Focused-ion beam sectioning**. Small amounts of separated samples were mounted on silicon wafers. After deposition of a carbon or tungstate protective layer, the microparticles on silicon wafers were cut to a thickness of <1 $\mu$m using a Ga-ion beam in an FIB apparatus (Hitachi SMI4050, Japan) and then transferred to Cu grids using a micromanipulator[58].

**Transmission electron microscopy**. FIB-cut microparticles were observed and analyzed using a JEM-ARM200F TEM equipped with an EDS (JEOL, Tokyo, Japan) operated at 200 kV. Microparticle mineral components were identified by the chemical compositions obtained via EDS analysis and the selected-area ED patterns in TEM[58].

**Inductively coupled plasma-mass spectrometry**. The separated samples were decomposed in 1 mL of 6 M $HNO_3$ and 0.2 mL of $H_2O_2$, dried, and then dissolved in 3 mL of 3 M $HNO_3$. High purity reagents (TAMAPURE AA-100 grade, Tama Chemical, Japan) were used to minimize the chemical blank. Next, a 2 mL aliquot of the sample solution was dried and dissolved in 0.15 M $HNO_3$ containing 10 ppb indium. The abundances of major and trace elements were measured using an Agilent 7700 ICP-MS, in which indium served as an internal standard. Isobaric interferences of Ba, Nd, Sm, Eu, Gd, and Tb with Eu, Gd, Tb, Dy, Ho, Er, Tm, Yb, and Lu were corrected using the corresponding elemental oxide production ratios $(MO^+/M^+)$ measured before and after analysis. All of the chemical analyses were performed in a clean room environment. To confirm the elemental compositions of the Mn-microparticles, we also conducted relative element composition analyses on separated microparticles ($n = 100$) using SEM-EDS. The results showed that, in the separated samples, only the Mn-microparticles contained manganese; thus ICP-MS manganese data were used to standardize and calculate the mass of other elements in the ICP-MS data. We also calculated the proportions of major Mn-microparticle elements (Mg, Al, K, Ca, Ti, Mn, and Fe) in the bulk sediment samples[21]. The REE values were plotted normalized to the Post-Archean Australian Shale (PAAS) values[59]. The Ce anomaly (Ce/Ce*) was also calculated by defining $(Ce/Ce^*) = Ce_N/(La^{1/2}Pr^{1/2})_N$[60], where the subscript $N$ denotes the PAAS-normalized abundance.

**XAFS spectroscopy**. To determine the major redox state of manganese and iron in the bulk sediment sample, Mn K-edge and Fe K-edge X-ray absorption fine structure (XAFS) spectra were measured at the Photon Factory, High Energy Accelerator Research Organization (KEK-PF, Tsukuba, Japan)[6]. The spectra were collected in the fluorescence yield mode using a 19-element solid state detector,

except for reference materials, which were measured in transmission mode. The X-ray absorption near edge structure (XANES) for manganese and iron and EXAFS for manganese were normalized and analyzed using REX2000 XAFS analysis software (Rigaku, Japan).

**Synchrotron-based STXM**. To determine the redox state of the Mn-microparticles, FIB-cut Mn-microparticles were also subjected to near-edge X-ray absorption fine structure (NEXAFS) spectra measurement at the Mn L-edge and Fe L-edge using an STXM apparatus at KEK-PF[47]. For the STXM reference spectra measurement, the powdered reagent was deposited on a Formvar-coated Cu grid. Also, to examine redox changes during repeated Mn(IV) reference (i.e., $\delta$-$MnO_2$) measurements at the Mn L-edge, sequences of transmission X-ray images (image stacks) were collected at photon energies spanning the relevant absorption edges (630–650 eV for the Mn L-edge and 700–720 eV for the Fe L-edge). The image stacks were aligned via spatial cross-correlation analysis, and NEXAFS spectra were obtained using IDL aXis2000 software (Hitchcock, and IDL-based analytical package, http://unicorn.mcmaster.ca).

**Thermodynamic calculations**. To estimate the saturation conditions under which manganese minerals precipitate, thermodynamic calculations were conducted for Mn(IV) and Mn(II) minerals using Visual MINTEQ 3.0 software[61]. We used the porewater chemical concentrations measured onboard[19] in these calculations.

## Data availability
Data supporting the findings of this study are included in this published Article and its Supplementary Information files.

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

## Acknowledgements

The samples used in this study were collected during IODP Expeditions 317 and 329 of the JOIDES Resolution and Japan Agency for Marine–Earth Science and Technology (JAMSTEC) cruise KY11-E06 aboard the R/V *Kaiyo*. We deeply appreciate the work of the shipboard crews, operational team members, and shipboard scientists who collected the sediment core samples. We would like to thank Drs. S. D'Hondt, R. Murray, A. Dunlea, A. Usui, and Y. Suzuki for the useful discussions, and T. Terada for his technical assistance in the laboratory. The synchrotron-based analyses were performed with the approval of the Japan Synchrotron Radiation Research Institute (2014A1239, 2014B1367, 2014B1372, 2015B1168, 2015B1168, 2015B1218, 2015B1487, 2016A1339, 2017A1103, 2017A1104, 2017B1239, 2017A1240, 2018A1419, and 2018A1421) and the High Energy Accelerator Research Organization (2013S2-003, 2016S-002, and 16G632). This study was supported in part by the Japan Society for the Promotion of Science (JSPS) Strategic Fund for Strengthening Leading-Edge Research and Development (to JAMSTEC and F.I.), the JSPS Funding Program for Next Generation World-Leading Researchers (GR102 to F.I.), JSPS Grant-in-Aid for Scientific Research (24687004 and 15H05608 to Y.M., 25871219 to G.-I.U., 15H02810 to R.W., 18H04134, 17H06458 and 17H04582 to Y.T., and 26251041 to F.I.), JSPS Grant-in-Aid for JSPS Fellows (14J00199 to G.-I.U.), and Ministry of Education, Culture, Sports, Science, and Technology (MEXT) Fund Leading Initiative for Excellent Young Researchers (to Kochi University and G.-I.U.).

## Author contributions

G.-I.U., Y.M., and F.I. designed the study; G.-I.U. performed resin-embedding processing and SEM image analysis; G.-I.U. and R.W. performed density concentration processing; G.-I.U., Y.M., and N.T. performed FIB processing and TEM observations; G.-I.U. and Y.M. performed flow cytometry/particle sorting processing; S.W. performed ICP-MS analysis; G.-I.U., Y.M., K.U., A.T., M.H. and Y.S. performed imaging via X-ray μCT; G.-I.U., Y.M., S.M., H.S., Ya.T. and Yo.T. performed STXM analysis; G.-I.U. and F.S.

compiled published bulk sediment composition data; G.-I.U., Y.M. and Yo.T. performed XAFS measurements; R.N. and Yo.T. performed XANES and EXAFS spectral analysis; R.N. conducted thermodynamic calculations; and G.-I.U., Y.M. and F.I. were the primary authors of the manuscript, with input from all other co-authors.

## Additional information

**Competing interests:** The authors declare no competing interests.

