## [Peer Review File · Nature Communications]

Reviewers' comments:

Reviewer #1 (Remarks to the Author):

Uramoto and co-workers report and quantify the amount of manganese microparticles present in the oxic abyssal sediments of the South Pacific Gyre (SPG) drilled during IODP expedition 329 in 2010. They conclude that the abundant Mn microparticles found in these deep-sea deposits were formed below the CCD through the supply of hydrothermal plumes. The authors have calculated and extrapolated that the amount of Mn present in the form of these microparticles is at least two orders of magnitude larger than the Mn budget of ferromanganese deposits found at the sediment surface.

The authors have produced a fantastic data set and I find the results of this study very interesting and of broad scientific significance and relevance. In particular, it is exciting to see that these deep and extremely remote seafloor environments which have remained understudied for so long offer so many new insights into geochemical and biogeochemical processes and fluxes occurring over geological time scales. However, I regret to say that in parts the manuscript appears immature and contains numerous imprecise statements. I also do not agree with some of the explanations and conclusions – at least was not convinced by the discussion and argumentation. Moreover, the English needs to be polished and several important references are missing. I can therefore not recommend publication of this manuscript in its current form.

Please find my major points, suggestions and specific comments below.

1) The discussion of early diagenetic cycling of manganese in marine sediments is insufficient and many key papers in this context have not been cited. In contrast to what the authors say, Mn-rich layers are common and typical features of deeper marine sediments in almost all ocean areas. Also a much more detailed discussion is needed to explain the liberation of Mn²⁺ in the investigated oxic sediments. The dissolved Mn²⁺ found in the pore waters of the investigated sediments demonstrate that reductive dissolution of Mn mineral phases occurs in the deeper parts of these sediments – accompanied by re-precipitation of secondary Mn oxides/hydroxides close to the sediment surface.

2) In this context also changes in bottom water and surface sediment redox over time and the significance for the diagenetic redistribution of Mn within the investigated sediments have not been considered adequately. The authors even state that they have found Mn(II) associated with carbonates which is a typical authigenic Mn mineral forming under suboxic/anoxic conditions. How and based on which evidence precisely do you know or suggest that the abundant Mn microparticles are primary mineral phases introduced into the sediment from hydrothermal plumes? Could it be possible that these microparticles – or part of it – originate from previous redox cycling of Mn (i.e. re-oxidation of Mn²⁺) within the sediments? There is growing evidence that the deep Pacific Ocean (and other ocean areas) experienced profound changes in bottom water ventilation and redox (and possibly also changes in the flux of organic matter to the seafloor) over glacial/interglacial timescales. A very interesting study in this respect is the one by Wegorzewski and Kuhn (2014) who show that a substantial part of manganese nodules collected from the Clarion Clipperton Zone in the equatorial Pacific Ocean formed by suboxic diagenesis – most likely during glacial periods when redox zonation within the surface sediments was more compressed than at present.

3) I am not convinced that the large amount of Mn microparticles found (entirely) originate from hydrothermal input. Why should hydrothermal plumes and thus the formation of Mn microparticles (in the water column) be restricted to below the CCD? The paper by Resing et al. (2015) clearly demonstrates that the hydrothermal plume reaches to much shallower water depth – i.e. up to about 2000 m). Hydrothermally formed ferromanganese deposits (for example on seamounts – ferromanganese crusts) tend to have specific Mn/Fe ratios and trace element composition. What are the Fe/Mn ratios of your microparticles? I assume that if the Mn microparticles were derived

from hydrothermal sources they should have specific Mn/Fe ratios and also specific trace element patterns that should clearly differ from oxic or suboxic authigenic Mn precipitates. Perhaps the authors may want to check this (e.g. review by Glasby (2006) which is cited in the manuscript).

Specific comments

L. 2: What do you mean with „dissolution“ in this context? Do you suggest that the ferromanganese concretions found at the sediment surface/on the seafloor are subject to dissolution? Do you mean ferromanganese „nodules“?

In the global element „cycle“ ? The cycle of which element precisely do you mean here?

Ls. 3 ff: Do you mean the extent of manganese nodules buried within the sediment? ... and do you refer to „formation“ pathways of nodules at the sediment surface or to buried nodules mentioned in this sentence? Please, be more precise.

L.12: It is not clear to me how and why the formation of the observed Mn microparticles should be restricted to the water masses/depth below the CCD.

Why „were“ hydrothermal plumes widespread? They are still widespread today (c.f. Resing et al., 2015).

A look at dissolved Fe and Mn concentrations in the water column around active hydrothermal vent sites which have become available within the framework of the GEOTRACES program demonstrate that hydrothermal plumes are in no way restricted to water depths below the CCD but well extent to shallower water depths – as shallow as 2.000 m – c.f. Resing et al. (2015). Why should the formation of the microparticles observed within the oxic sediments of SPG then be restricted to water depths below the CCD? This is not clear and convincing to me at all.

Ls. 27 ff: I do not agree with this statement at all. There are several papers that have shown that Mn nodules preserved/buried in deep-sea sediments are found at numerous locations. (e.g., von Stackelberg, 1987; Mewes et al., 2014) Moreover, Mn-rich layers are typical features of marine sediments and widespread in all ocean areas. While authigenic Mn oxides/hydroxides typically form at the active oxic-anoxic boundary, these Mn-rich layers are often converted into and preserved as Mn carbonates during burial into suboxic/anoxic sediments (e.g., Neumann et al., 2002). Please also check – amongst others - the contributions by Burdige and Gieskes(1983) and Burdige (1993, 2006) on the early diagenetic cycling of manganese in marine sediments.

L. 35: needs to be „These“ instead of This.

L. 36: “represent” instead of “represents”

L. 41: The authors state that a „newly“ developed resin embedding technique has been used. To my knowledge such techniques have been developed and presented previously (e.g., Jilbert et al., 2008).

What exactly do you mean with „downstream“ analyses? This is not clear to me.

L. 64: it has to be „oxic“ instead of „aerobic“

How do you know that the redox conditions in the SPG were stable – or more precisely oxic - over the whole time of deposition of the sediments? There are numerous studies that demonstrate that the oxygen and redox conditions in the abyssal Pacific Ocean water masses experienced significant changes in oxygen concentrations. Therefore, I am not surprised that Mn(II) is found within the deeper sediments (I suspect that it is mostly Mn carbonate) suggesting that the sediments of the SPG also experienced suboxic conditions in the past.

L. 70: I do not understand this part of the sentence. What exactly do you mean with “inorganic

carbon might be transformed to the mineral phase during the burial" ? Which mineral phase do you mean precisely and how precisely do you think is the carbon "transformed"?

L. 71: Do you mean "redox changes induced by microbial activity"? Which kind of microbial activity do you refer to?

L. 87: relative "to" total Mn in the bulk sediment

Ls. 88 ff: I do not understand your line of argumentation in this sentence. Please rephrase.

Ls. 92/93 ff. and figures 2 and 5 (extended data): I find it hard to follow the approach and interpretation you are presenting here. What is the use of plotting the number of Mn microparticles in the sediments (partly of deep subsurface sediments) versus dissolved Mn and Fe concentrations in the modern bottom water? In figures 2b and 5b you speak of surface sediments. What are these "surface sediments"? I know that during drilling – and even during coring by gravity or piston corer - you lose the uppermost sediments. So, I assume that what you mean here is the "top of the core" and not the surface sediment proper, right? If yes, please say so. In this respect I also do not understand why you should plot the abundance of Mn particles in sediments which are tens or hundreds of thousands of years older than the current bottom water mass?!

L. 98: What do you mean with "aquagenic"? Which process are you referring to here?

Ls. 103 ff (and fig. 7c extended data): I am sorry, but in this cloud of symbols any depth trend is really hard to tell. I see slightly elevated contents in the lower portion of the cores – maybe these indicate the depth of active manganese reduction and could also explain why you observe the decrease in Mn content of the Mn microparticles with increasing depth because here they have already been reductively dissolved). How do you explain that Mn reduction occurs in these (still) oxic parts of the sediments? Is it possible that the ex situ determined oxygen concentrations are higher than those in situ due to the consolidated nature of the deep sediments?

Ls. 113-115: Of course! Fine-grained Mn oxide particles are definitely more susceptible to reduction than Mn contained in clay minerals.

Ls. 115-117: Which "site" precisely do you refer to here? The argumentation in this last part of the sentence is absurd. The sampling location/s has/have definitely not migrated tectonically underneath water masses. Water masses are not immobile water bodies which "sit" in the ocean basins But in contrast the changes in the flow and extent of ventilation of deep water masses may most likely have induced changes in redox condition in the surface sediments over glacial/interglacial timescales. These are definitely much faster than tectonic movement of oceanic crust. Moreover you suddenly speak of anoxic sediments while before you stated that the SPG sediments are oxic throughout?! This is contradictory and hard to follow.

Ls. 127 ff: The wording here is a bit strange – or let's say contradictory. How can these Mn microparticles contribute significantly to element cycling if they are stable in the seafloor?!

Additional references and recommended reading

Burdige (1993) *Earth Sci. Rev.*, 35,249-284.

Burdige (2006) *Geochemistry of Marine Sediments*, Princeton.

Burdige and Gieskes (1983) *Am. J. Sci.*, 283, 29-47.

Jilbert et al. (2008), *L & O Methods*, 6, 16-22.

Mewes et al. (2014), *Deep-Sea Res. I*, 91,125-141.

Neumann et al. (2002), *GCA*, 66, 5,867-879.

Von Stackelberg (1987) *Pumice and buried manganese nodules from the equatorial North Pacific Ocean. Geol. Jb.*, 3-25.

Wegorzewski and Kuhn (2014), *Mar. Geol.*, 357, 123-141.

Reviewer #2 (Remarks to the Author):

The study by Uramoto et al. yields novel and excitingly unexpected results about the fate and distribution of manganese in the oceans. Through a rigorous analysis of the geochemistry and mineralogy of oxic, deep sea sediments that occur throughout the South Pacific Gyre, these results demonstrate that manganese is much more abundant in deep sea sediments than ever accounted for – increasing the global Mn budget, and particularly the budget of the deep sea. The manganese is largely contained in Mn-rich microparticles containing trace amounts of organics and carbonates, but interestingly, the Mn oxidation state is Mn(II) in these microparticles despite the oxic environment. The Mn-rich particles are present throughout the South Pacific Gyre in oxic pelagic clays ranging the seafloor to the basement in multiple drillcores. These results greatly expand our understanding of Mn geochemical cycling in the deep ocean, as previously it was thought that Mn existed largely as Mn(III/IV) oxide minerals in Mn nodules and pavements primarily at the seafloor. Not only does this information regarding the abundance and distribution of these Mn microparticles change our perception about the global Mn budget, but the reduced oxidation state (Mn(II)) requires us to re-examine our understanding the impact of Mn biogeochemistry and, more broadly, deep ocean processes

The science will be of interest to the entire oceanographic community and I will have broader interest for those studying early earth processes, as Mn-rich carbonates are prevalent right during the Great Oxygenation Event (GOE). The scientific approach and methodology is very rigorous, and all of the figures (both within the main manuscript and supplemental information) are clear and valuable.

I think this study should be published in Nature Communications, but I do have a few recommendations to strengthen the article:

1. It is noted that there still is some Mn present throughout the sediment that is more finely disseminated (in other words, not associated with these Mn-rich microparticles). It would be really valuable to know what the phase and oxidation state of this Mn is, perhaps through bulk EXAFS of the sediment with the microparticles sieved out. While this Mn in the bulk sediment is not concentrated, it could be important to know whether this Mn is also present as a Mn(II)-rich phase, or if these are present as an oxidized Mn(III/IV) phase, particularly with depth into the sediment column. Even if the bulk sediment doesn't contribute as much to the Mn budget, it would be valuable for interpretation of Mn geochemical cycling and formation/preservation of Mn(II) minerals in these oxic environments.
2. The rationale for the inverse relationship between Mn-particle concentration and bottom seawater Mn concentration is not clear to me. It would be useful to explain to the reader the mechanisms by which this relationship could be explained.
3. I don't entirely understand the statement on line 113 "Since Mn microparticles are absent and clay-microparticles are abundant, our findings suggest that Mn microparticles are more redox sensitive...". If these particles already contain reduced Mn, why would they be more redox sensitive when transitioned to an anoxic environment? Also, why don't Mn(II) carbonate-containing microparticles form in anoxic environments? Is there a mechanism to explain this? I think this general argument needs to be better clarified.
4. It would be useful to show some linear combination fitting results of the NEXAFS data in Figure 3 to show what proportion of reference spectra represents each spectra. Also, is there a reference for Mn(II) sorbed onto carbonates? This would be useful to include as a reference.

Reviewer #3 (Remarks to the Author):

In this work, Uramoto et al. do demonstrate that there is a remarkable abundance of Mn microparticles that have been deposited and preserved in the massive amount of sediments that have slowly accumulated in the highly oligotrophic South Pacific Gyre. This is the key observation in the manuscript. At the large scale, the authors then try to extrapolate their data to other oxic sediments to create a sense of scale for the potential amount of subsurface Mn that is not currently well accounted for in global budget. At the small scale (i.e. scale of particles), the authors show that there is some organic carbon associated with the Fe/Mn particles, and they also try to examine the potential mineralogy and formation mechanism.

It is truly surprising that manganese microparticles are preserved up to 100 million years in subseafloor sediments. The extent to which these particles are primary vs. secondary (i.e. have been modified by diagenetic alteration) is not always clear. At times the authors refer to the microparticles as Fe/Mn particles, presumably mixtures of Fe and Mn oxides that are in those tangles with the organic carbon. It is inferred that these particles form from oxidation and deposition of Fe(II) and Mn(II) dispersed from hydrothermal systems in the deep ocean. Yet papers on this topic, which also include x-ray spectromicroscopy characterization of such phases, such as works by Toner et al. on the formation and preservation of carbon-associated Fe-particles and their long distance transport from hydrothermal sites, or such as by Templeton et al. in the formation of Fe/Mn oxides and carbon that get deposited onto rock and sediment surfaces in the deep ocean away from direct hydrothermal input, are not referenced. Some of the microparticle images in Figure 1 in this work even look v. similar to SEM and FIB/TEM images of Fe/Mn-organic phases embedded into basalt surfaces observed in Templeton et al., (2009). Thus it seems that this work should be placed within some of the processes examined by those authors.

In this work, the authors have made efforts to specifically characterize the Mn and carbon speciation. The Mn appears to be MnCO₃, which is likely not primary but rather a secondary product (i.e. from Mn(IV)-reduction). This is mentioned briefly but not necessarily clearly articulated for the reader. Also, is there spectroscopic data of some of the particles that shows Mn(IV), for example in the Fe/Mn rich materials? It would have been very helpful to have bulk Mn XANES for many samples from this study in order to get an average Mn-oxidation state. Instead, we only have very limited data for a few particles that were subjected to Mn L-edge NEXAFS analysis during STXM imaging.

For the Fe L-edge, why do the authors not try to determine Fe oxidation state? Are these particles primarily Fe(II), Fe(III) or a mixture? A few model compounds are shown, but not enough for a reader to draw a conclusion. There is some excellent data about Fe(II/III) quantification from Fe L-edge XANES in Bourdelle et al. (2013). I expect some of those Toner papers also contain good comparisons.

Another question about the spectroscopic data. It seems that there are Mn NEXAFS for ABCDEF (three spots, two different particles), and C XANES for ABC (3 spots on just one particle). Given the scope of the samples assessed in this study, why isn't there a much larger data set? It seems like many more particles should have been imaged, giving rise to many more Mn and C spectra for statistical analysis and an assessment of variability in forms of Mn and C that are present.

Although the Mn NEXAFS have very good signal to noise and are clearly comparable to the model shown (Mn-carbonate), the C NEXAFS data is quite noisy and hard to compare against models. It would be good if the authors at least pointed out major transitions in the data and showed more clearly how those transitions fit against the model compounds (they invoke humics and fulvics as the best fit). It is very common in such C NEXAFS analysis to identify the actual bonds probed and to point out the well-calibrated energy position of aliphatics, amides, carbonates, etc. Examples can be found in papers by Clara Chan, Julie Cosmidis and many others.

For the electron diffraction that is presented, is the data being put into a useful framework for proving this is a FeMn mineral? Or are the authors only using the EDS data? Please clarify.

The authors do possess measured porewater alkalinity, DIC and Mn profiles, so it would be useful if they would also provide calculations of where MnCO_3 may be supersaturated at various depths.

The authors mention that there are corresponding microbial data sets. Have they looked into this data to derive any sense if Mn-oxidizers (or reducers) may be present in the sediment as well?

There is a huge growth in studies right now that are focused on how Fe and Mn oxides can stabilize organic matter and lead to its preservation, mostly in the soil science vs. oceanography literature. Since the authors are surprised by this phenomenon in their data set, due to the oligotrophic conditions of the South Pacific Gyre, it would be good for them to research this further.

The manuscript would also benefit from a conceptual model. It would be good to create more of a timeline regarding processes the authors invoke: loading of Fe/Mn-organic tangles in to these sediments, with the silicate fraction, followed by in-situ transformations that occur (that should be specifically defined), and fate of the microparticles with time and the Fe, Mn and C components they contain (e.g. formation of MnCO_3 as one example). Right now there is too much ambiguity and many readers could misinterpret this study.

Response to the reviewers' comments:

Reviewer #1:

Uramoto and co-workers report and quantify the amount of manganese microparticles present in the oxic abyssal sediments of the South Pacific Gyre (SPG) drilled during IODP expedition 329 in 2010. They conclude that the abundant Mn microparticles found in these deep-sea deposits were formed below the CCD through the supply of hydrothermal plumes. The authors have calculated and extrapolated that the amount of Mn present in the form of these microparticles is at least two orders of magnitude larger than the Mn budget of ferromanganese deposits found at the sediment surface.

The authors have produced a fantastic data set and I find the results of this study very interesting and of broad scientific significance and relevance. In particular, it is exciting to see that these deep and extremely remote seafloor environments which have remained understudied for so long offer so many new insights into geochemical and biogeochemical processes and fluxes occurring over geological time scales. However, I regret to say that in parts the manuscript appears immature and contains numerous imprecise statements. I also do not agree with some of the explanations and conclusions – at least was not convinced by the discussion and argumentation. Moreover, the English needs to be polished and several important references are missing. I can therefore not recommend publication of this manuscript in its current form.

Please find my major points, suggestions and specific comments below.

We thank Reviewer 1 for providing thoughtful comments and suggestions, which significantly improved the manuscript. We have addressed all of your comments and suggestions as our responses below.

Comment #1.1

The discussion of early diagenetic cycling of manganese in marine sediments is insufficient and many key papers in this context have not been cited. In contrast to what the authors say, Mn-rich layers are common and typical features of deeper marine sediments in almost all ocean areas.

The authors appreciate this comment. In the revised manuscript, we added some important references for the occurrence of manganese in a variety of marine sediments; however, to our best knowledge, the nature of manganese in entirely oxic deep-subseafloor sediments remains poorly unknown. To avoid confusion for the readers, we described the novelty of our findings more clearly by addressing your comments and suggestions in the revised manuscript.

Comment #1.2

Also a much more detailed discussion is needed to explain the liberation of Mn²⁺ in the investigated oxic sediments. The dissolved Mn²⁺ found in the pore waters of the investigated sediments demonstrate that reductive dissolution of Mn mineral phases occurs in the deeper parts of these sediments – accompanied by re-precipitation of secondary Mn oxides/hydroxides close to the sediment surface.

To address this point properly, we re-assessed the redox state of manganese by XAFS analysis of the bulk sediment, and also performed thermodynamic calculation on the sediment. In fact, XAFS results clearly verify the majority of manganese was Mn(IV) (Fig. 5). Thermodynamic calculation indicates that Mn(IV) are saturated, whereas Mn(II) are unsaturated in oxic subseafloor environment. In addition, the STXM

spectra of manganese showed experimental photoreduction of manganese by intense X-ray irradiation; in this regard, we estimated photon flux by relevant references in XAFS and STXM analyses and discussed the possibility of photoreduction in STXM measurements (lines 144-167). Given those new analysis and re-examination of mineralogical data, we concluded that the redox state of Mn-microparticles is Mn(IV). Accordingly, we made additional discussion on the formation process of Mn(IV)-microparticles at lines 127-143 in the revised version.

Comment #1.3

In this context also changes in bottom water and surface sediment redox over time and the significance for the diagenetic redistribution of Mn within the investigated sediments have not been considered adequately. The authors even state that they have found Mn(II) associated with carbonates which is a typical authigenic Mn mineral forming under suboxic/anoxic conditions.

We appreciate this important comment. The redox state of the bottom water could be an important factor for the formation of manganese oxide in sediment. We additionally described this point at lines 131-133 and added a reference. The authors also agree that Mn(II)-associated carbonates can be seen in suboxic/anoxic sediments on the Pacific margins; we added the references, accordingly.

Comment #1.4

How and based on which evidence precisely do you know or suggest that the abundant Mn microparticles are primary mineral phases introduced into the sediment from hydrothermal plumes?

This is a very important comment, thank you. Given the recent observations in the GEOTRACE, hydrothermal plumes generally occur in relatively shallow water column above CCD. We plotted Fe-Mn-(Ni + Cu)*10 concentrations in Mn-microparticles onto the ternary diagram, showing that these compositions are not tightly associated with the trend of hydrothermally produced manganese (Fig. 4e). Consequently, we removed our previous statements regarding this hydrothermal scenario in the revised manuscript.

Comment #1.5

Could it be possible that these microparticles – or part of it – originate from previous redox cycling of Mn (i.e. re-oxidation of Mn²⁺) within the sediments? There is growing evidence that the deep Pacific Ocean (and other ocean areas) experienced profound changes in bottom water ventilation and redox (and possibly also changes in the flux of organic matter to the seafloor) over glacial/interglacial timescales. A very interesting study in this respect is the one by Wegorzewski and Kuhn (2014) who show that a substantial part of manganese nodules collected from the Clarion Clipperton Zone in the equatorial Pacific Ocean formed by suboxic diagenesis – most likely during glacial periods when redox zonation within the surface sediments was more compressed than at present.

Thanks for this comment and references. Although multiple lines of additional analytical evidences showed that redox state of manganese in Mn-microparticle is Mn(IV), we introduced the interesting study of manganese nodules at lines 16-29 in the revision text of Introduction.

Comment #1.6

3) I am not convinced that the large amount of Mn microparticles found (entirely) originate from hydrothermal input. Why should hydrothermal plumes and thus the formation of Mn microparticles (in the water column) be restricted to below the CCD? The paper by Resing et al. (2015) clearly demonstrates that the hydrothermal plume reaches to much shallower water depth – i.e. up to about 2000 m). Hydrothermally formed ferromanganese deposits (for example on seamounts – ferromanganese crusts) tend to have specific Mn/Fe ratios and trace element composition. What are the Fe/Mn ratios of your microparticles? I assume that if the Mn microparticles were derived from hydrothermal sources they should have specific Mn/Fe ratios and also specific trace element patterns that should clearly differ from oxic or suboxic authigenic Mn precipitates. Perhaps the authors may want to check this (e.g. review by Glasby (2006) which is cited in the manuscript).

Thanks again for the useful information and comments. We deleted the hydrothermal story in the revised version (see #1.4). We showed Fe-Mn-(Ni + Cu)*10 ratios in new Fig. 4e (also see our response to #1,4).

Comment #1.7

L. 2: What do you mean with „dissolution“ in this context? Do you suggest that the ferromanganese concretions found at the sediment surface/on the seafloor are subject to dissolution? Do you mean ferromanganese „nodules“? In the global element „cycle“ ? The cycle of which element precisely do you mean here?

We rephrased the sentence as follows: “Manganese minerals are widely distributed on the seafloor of the global abyssal plains, and the formation and diagenetic processes of these minerals are important for better understanding the global marine manganese cycle.” (lines 1-3).

Comment #1.8

Ls. 3 ff: Do you mean the extent of manganese nodules buried within the sediment? ... and do you refer to „formation“ pathways of nodules at the sediment surface or to buried nodules mentioned in this sentence? Please, be more precise.

We rephrased the sentence as follows: “However, the extent of ferromanganese minerals buried within entirely oxic subseafloor sediments remains unclear.” (lines 3-4).

Comment #1.9

L.12: It is not clear to me how and why the formation of the observed Mn microparticles should be restricted to the water masses/depth below the CCD.

This is based on the fact that we could not observe Mn-microparticles in calcareous oozes (lines 79-80, Fig. 3a), and also limited Mn-oxides have been detected in the sediments above the CCD where the sedimentation rate is generally higher than that of abyssal plain below the CCD (Hein et al., 2013, *Ore Geol. Rev.* **51**, 1–14) (lines 127-143).

Comment #1.10

Why „were“ hydrothermal plumes widespread? They are still widespread today (c.f. Resing et al., 2015).

A look at dissolved Fe and Mn concentrations in the water column around active hydrothermal vent sites which have become available within the framework of the GEOTRACES program demonstrate that hydrothermal plumes are in no way restricted to water depths below the CCD but well extent to shallower water depths – as shallow as

2.000 m – c.f. Resing et al. (2015). Why should the formation of the microparticles observed within the oxic sediments of SPG then be restricted to water depths below the CCD? This is not clear and convincing to me at all.

Because the Fe-Mn-(Ni + Cu)*10 diagram for Mn-microparticles did not support the notion of the hydrothermal origin (Fig. 4e) and also hydrothermal plumes well extent to the shallower water depths than the CCD, we concluded that those microparticles were not directly originated from the plumes. In the revised manuscript, we focused on the formation and diagenetic processes in the subseafloor sediment (lines 127-143 and 168-186, also see our responses to #1.4 and #1.9)

Comment #1.11

Ls. 27 ff: I do not agree with this statement at all. There are several papers that have shown that Mn nodules preserved/buried in deep-sea sediments are found at numerous locations. (e.g., von Stackelberg, 1987; Mewes et al., 2014) Moreover, Mn-rich layers are typical features of marine sediments and widespread in all ocean areas. While authigenic Mn oxides/hydroxides typically form at the active oxic-anoxic boundary, these Mn-rich layers are often converted into and preserved as Mn carbonates during burial into suboxic/anoxic sediments (e.g., Neumann et al., 2002). Please also check – amongst others - the contributions by Burdige and Gieskes(1983) and Burdige (1993, 2006) on the early diagenetic cycling of manganese in marine sediments.

According to your comments, we described the distribution of manganese in suboxic-to-anoxic marine sediments with multiple representative references at lines 16-29. By addressing this point, we could improve the clarity of our statements.

Comment #1.12

L. 35: needs to be „These“ instead of This.

Corrected.

Comment #1.13

L. 36: “represent” instead of “represents”

Corrected.

Comment #1.14

L. 41: The authors state that a „newly“ developed resin embedding technique has been used. To my knowledge such techniques have been developed and presented previously (e.g., Jilbert et al., 2008).

What exactly do you mean with „downstream“ analyses? This is not clear to me.

In this study, we used resin-embedding technique in which a biological technique was applied to retain fine texture and arrangement of mineral grains and organic materials in sediments. Since this was already published (Uramoto et al. 2014), we removed the term of “newly” from this sentence, and provided more detailed description for the method and its importance in this study (lines 40-42).

Comment #1.15

L. 64: it has to be „oxic“ instead of „aerobic“

Corrected.

Comment #1.16

How do you know that the redox conditions in the SPG were stable – or more precisely oxic - over the whole time of deposition of the sediments? There are numerous studies that demonstrate that the oxygen and redox conditions in the abyssal Pacific Ocean water masses experienced significant changes in oxygen concentrations. Therefore, I am not surprised that Mn(II) is found within the deeper sediments (I suspect that it is mostly Mn carbonate) suggesting that the sediments of the SPG also experienced suboxic conditions in the past.

An important fact is that the concentrations of Mn-microparticles are generally constant in the entirely oxic SPG sediment columns (also see Fig. 2b in D'Hondt et al., *Nat. Geosci.*, 2015); nevertheless, the authors agree that we could not get rid of the possibility of paleoceanographic redox changes in the bottom seawater. In the revised manuscript, we described these multiple possibilities (Also see our response to #1.5.).

As for the occurrence of Mn(II) by STXM, we additionally analyzed the bulk sediment with XAFS. Also, we conducted thermodynamic calculation of pore water and re-examined mineralogical data. These additional analysis and data systematically indicate that Mn(IV) consists of manganese in Mn-microparticles (see lines 144-167; Figs. 2, 4, 5).

Comment #1.17

L. 70: I do not understand this part of the sentence. What exactly do you mean with “inorganic carbon might be transformed to the mineral phase during the burial” ? Which mineral phase do you mean precisely and how precisely do you think is the carbon “transformed”?

In the original version, we intended to describe the occurrence of Mn(II)-carbonates in the microparticles as an initial diagenetic process in the SPG sediment. However, as described above, we confirmed that the mineral phase of manganese in Mn-microparticles is Mn(IV) and therefore we deleted this statement in the revised manuscript.

Comment #1.18

L. 71: Do you mean “redox changes induced by microbial activity”? Which kind of microbial activity do you refer to?

This implicates the occurrence of Mn-reducers associated with manganese minerals. There are interesting data sets showing biologically-driven manganese redox cycling inside the manganese nodules from northeastern equatorial pacific (Blöthe e al. 2015). Also, Tully and Heidelberg (2013) found that manganese nodules at SPG also harbored similar microbial community. However, both studies detected Mn-specific microbial community only in manganese nodules that is distinct from surrounding sediments. Also, microbial community structure in deeper sediment have not been examined because of very low biomass present in the sediments (D'Hondt et al. 2015). Although it is very interesting, the involvement of microbial activity in diagenetic dissolution of Mn-microparticles is too speculative and we deleted the corresponding sentence in the revised manuscript.

Comment #1.19

L. 87: relative “to” total Mn in the bulk sediment

Corrected.

Comment #1.20

Ls. 88 ff: I do not understand your line of argumentation in this sentence. Please rephrase.

We rephrased the sentence as: “The amount of manganese in Mn-microparticles is particularly high, accounting for an average of 42% of the total manganese in bulk SPG sediment samples²⁴ (Fig. 4 a–d). The percentage is high at shallow depths (>30–60%), but the mass of manganese in Mn-microparticles relative to the total manganese of bulk sediments shows an overall decrease with increasing depth (<0.1% of the total manganese in sediments; see Fig. 4d)” (lines 89-93).

Comment #1.21

Ls. 92/93 ff. and figures 2 and 5 (extended data): I find it hard to follow the approach and interpretation you are presenting here. What is the use of plotting the number of Mn microparticles in the sediments (partly of deep subsurface sediments) versus dissolved Mn and Fe concentrations in the modern bottom water? In figures 2b and 5b you speak of surface sediments. What are these “surface sediments”? I know that during drilling – and even during coring by gravity or piston corer - you lose the uppermost sediments. So, I assume that what you mean here is the “top of the core” and not the surface sediment proper, right? If yes, please say so. In this respect I also do not understand why you should plot the abundance of Mn particles in sediments which are tens or hundreds of thousands of years older than the current bottom water mass?!

The authors generally agree with you. The samples we plotted as “surface sediments” are not exactly on the seafloor (i.e., modern sediment). Those sediment samples have already aged at least over 0.5 Ma, which are not simply comparable to the modern bottom water-seafloor processes. Accordingly, we omitted the previous Fig. 2b and Extended Data Fig. 5b in the revised version.

Comment #1.22

L. 98: What do you mean with “aquagenic”? Which process are you referring to here?

In the original manuscript, we meant the accumulation process of minerals (e.g. iron and manganese) in the water column related to hydrothermal processes. In the response to your previous comment (#1.4), we plotted Fe-Mn-(Ni + Cu)*10 in Mn-microparticles, which clearly showed that these compositions are not associated with the trend of hydrothermally produced manganese (Fig. 4e). Consequently, we removed our previous statements in the revised manuscript.

Comment #1.23

Ls. 103 ff (and fig. 7c extended data): I am sorry, but in this cloud of symbols any depth trend is really hard to tell. I see slightly elevated contents in the lower portion of the cores – maybe these indicate the depth of active manganese reduction and could also explain why you observe the decrease in Mn content of the Mn microparticles with increasing depth because here they have already been reductively dissolved). How do you explain that Mn reduction occurs in these (still) oxic parts of the sediments? Is it possible that the ex situ determined oxygen concentrations are higher than those in situ due to the consolidated nature of the deep sediments?

To gain the clarity, we replotted the previous Extended Data Fig. 7c and replaced it to Fig. S8a in the revised manuscript so that depth trend of porewater manganese is more clearly visible. In addition, we improved the discussion part on redox state of manganese in Mn-microparticles as Mn(IV) based on the results of TEM-EDS-ED, bulk XAFS, and thermodynamic calculation (Lines 144-167). We also added explanation for the experimental photoreduction of manganese in Mn-microparticles during STXM measurements at lines 144-167 in the revised Discussion section.

Comment #1.24

Ls. 113-115: Of course! Fine-grained Mn oxide particles are definitely more susceptible to reduction than Mn contained in clay minerals.

To make the redox state of manganese in the bulk sediment and particle inside clearer, we additionally performed XAFS of bulk samples, STXM of separated Mn-microparticles, thermodynamic calculation of sediment, and re-examine the TEM-EDS-ED of separated Mn-microparticles. The results show that redox state of manganese in Mn-microparticles is Mn(IV) and STXM spectra represent the occurrence of experimental photoreduction.

Comment #1.25

Ls. 115-117: Which “site” precisely do you refer to here? The argumentation in this last part of the sentence is absurd. The sampling location/s has/have definitely not migrated tectonically underneath water masses. Water masses are not immobile water bodies which “sit” in the ocean basins But in contrast the changes in the flow and extent of ventilation of deep water masses may most likely have induced changes in redox condition in the surface sediments over glacial/interglacial timescales. These are definitely much faster than tectonic movement of oceanic crust. Moreover you suddenly speak of anoxic sediments while before you stated that the SPG sediments are oxic throughout?! This is contradictory and hard to follow.

In the revised manuscript, we describe the possible diagenesis and redox sensitivity of Mn-microparticles within the oxic sedimentary environments. At lines 168-186, we deleted the statement of tectonics or anoxic sediment to gain the readability.

Comment #1.26

Ls. 127 ff: The wording here is a bit strange – or let’s say contradictory. How can these Mn microparticles contribute significantly to element cycling if they are stable in the seafloor?!

We rephrased the sentence as: “This discovery of the abundant Mn-microparticles provides new insights into the formation/preservation of ferromanganese minerals in deep subseafloor environments and improve our understanding of the global manganese budget and element cycle over geologic time.” (lines 197-199).

Additional references and recommended reading

Burdige (1993) Earth Sci. Rev., 35,249-284.

Burdige (2006) Geochemistry of Marine Sediments, Princeton.

Burdige and Gieskes (1983) Am. J. Sci., 283, 29-47.

Jilbert et al. (2008), L & O Methods, 6, 16-22.

Mewes et al. (2014), Deep-Sea Res. I, 91,125-141.

Neumann et al. (2002), GCA, 66, 5,867-879.

Von Stackelberg (1987) Pumice and buried manganese nodules from the equatorial North Pacific Ocean. Geol. Jb., 3-25.

Wegorzewski and Kuhn (2014), Mar. Geol., 357, 123-141.

We appreciate your recommendations, those references greatly helped us to revise this manuscript.

Reviewer #2 (Remarks to the Author):

The study by Uramoto et al. yields novel and excitingly unexpected results about the fate and distribution of manganese in the oceans. Through a rigorous analysis of the geochemistry and mineralogy of oxic, deep sea sediments that occur throughout the South Pacific Gyre, these results demonstrate that manganese is much more abundant in deep sea sediments than ever accounted for – increasing the global Mn budget, and particularly the budget of the deep sea. The manganese is largely contained in Mn-rich microparticles containing trace amounts of organics and carbonates, but interestingly, the Mn oxidation state is Mn(II) in these microparticles despite the oxic environment. The Mn-rich particles are present throughout the South Pacific Gyre in oxic pelagic clays ranging the seafloor to the basement in multiple drillcores. These results greatly expand our understanding of Mn geochemical cycling in the deep ocean, as previously it was thought that Mn existed largely as Mn(III/IV) oxide minerals in Mn nodules and pavements primarily at the seafloor. Not only does this information regarding the abundance and distribution of these Mn microparticles change our perception about the global Mn budget, but the reduced oxidation state (Mn(II)) requires us to re-examine our understanding the impact of Mn biogeochemistry and, more broadly, deep ocean processes

The science will be of interest to the entire oceanographic community and I will have broader interest for those studying early earth processes, as Mn-rich carbonates are prevalent right during the Great Oxygenation Event (GOE). The scientific approach and methodology is very rigorous, and all of the figures (both within the main manuscript and supplemental information) are clear and valuable.

I think this study should be published in Nature Communications, but I do have a few recommendations to strengthen the article:

The authors thank Reviewer 2 for your positive comments and useful suggestions. Addressing your points has indeed significantly contributed to the revised manuscript.

Comment #2.1

It is noted that there still is some Mn present throughout the sediment that is more finely disseminated (in other words, not associated with these Mn-rich microparticles). It would be really valuable to know what the phase and oxidation state of this Mn is, perhaps through bulk EXAFS of the sediment with the microparticles sieved out. While this Mn in the bulk sediment is not concentrated, it could be important to know whether this Mn is also present as a Mn(II)-rich phase, or if these are present as an oxidized Mn(III/IV) phase, particularly with depth into the sediment column. Even if the bulk sediment doesn't contribute as much to the Mn budget, it would be valuable for interpretation of Mn geochemical cycling and formation/preservation of Mn(II) minerals in these oxic environments.

The authors thank you for this critical point. To confirm the major redox state of manganese in the sediments, we additionally performed XAFS analyses using the bulk samples. The bulk analyses clearly showed that manganese consists mainly of Mn(IV) (Fig. 5). In addition, we performed thermodynamic calculation of pore water of samples and re-examined the mineral character, consistently suggesting that the manganese in Mn-microparticles is Mn(IV). Moreover, we repeated measurements for the Mn(IV) reference (i.e., δ -MnO₂) and found that photoreduction of Mn(IV) most likely occurred during the STXM

measurements. According to these multiple lines of additional evidences, we conclude that the redox state of manganese in Mn-microparticles is to be Mn(IV). SEM-EDS elemental mapping of manganese in the resin-embedded sample also showed that Mn-microparticles highly concentrated manganese in those sediments. We believe that these data well support our notion that manganese within microparticles play roles in the accumulation of manganese in the oxic sedimentary environments over geologic time (~100 million years).

Comment #2.2

The rationale for the inverse relationship between Mn-particle concentration and bottom seawater Mn concentration is not clear to me. It would be useful to explain to the reader the mechanisms by which this relationship could be explained.

Thanks to pointing out this. The samples we plotted as “surface sediments” in the pointed figures (previous Fig. 2b and Extended Data Fig. 5b) have already aged at least ~0.5 Ma, and therefore it cannot be simply comparable to the modern bottom water-seafloor processes. Accordingly, we omitted the previous Fig. 2b and Extended Data Fig. 5b in the revised manuscript.

Comment #2.3

I don't entirely understand the statement on line 113 “Since Mn microparticles are absent and clay-microparticles are abundant, our findings suggest that Mn microparticles are more redox sensitive...). If these particles already contain reduced Mn, why would they be more redox sensitive when transitioned to an anoxic environment? Also, why don't Mn(II) carbonate-containing microparticles form in anoxic environments? Is there a mechanism to explain this? I think this general argument needs to be better clarified.

In the revised manuscript, the reason for the redox-sensitive nature of Mn-microparticles was described based on the confirmation of redox state of manganese in Mn-microparticles during this revision. Multiple additional data (XAFS analysis of bulk sediments and thermodynamic calculation of pore water, and re-examination of mineralogical characters of Mn-microparticles) demonstrated that manganese in Mn-microparticles is Mn(IV) and supported our notion that the absence of manganese oxides in the anoxic sedimentary environment (lines 168-186).

Comment #2.4

It would be useful to show some linear combination fitting results of the NEXAFS data in Figure 3 to show what proportion of reference spectra represents each spectra. Also, is there a reference for Mn(II) sorbed onto carbonates? This would be useful to include as a reference.

To determine Mn-oxidation state of Mn-microparticles precisely, we conducted multiple examinations based on XAFS analysis of the bulk sediment samples, the calculation of thermodynamic properties of pore water of samples, and re-examination of data of mineral character (see, our response to #2.1). In addition, we conducted the repeated measurement of Mn(IV) reference mineral (δ -MnO₂) and found the changes in redox state of Mn(IV) during spectroscopic measurement. Taken together, we concluded that Mn-oxidation state in Mn-microparticles is Mn (IV), and Mn (II) signals on NEXAFS/STXM should be from experimental photoreduction during STXM measurement. This means, in the current state of STXM

analyses, precise determination of Mn redox state is technically difficult, and we carefully discussed these data interpretations (including the possibility of artificial photoreduction of manganese) in the Discussion part of the revised manuscript (lines 144-167).

Also, to address your point, we conducted adsorption experiment of Mn(II) onto carbonate based on Wersin et al. (1989, *Geochim. Cosmochim. Acta*, **53**, 2787–2796). However, we could not detect Mn(II) signal by STXM potentially due to extremely low amount of Mn(II) on carbonate. This experimental result agreed well with the prediction by thermodynamic calculation done for estimating the amount of absorbed Mn(II) on carbonate.

Reviewer #3 (Remarks to the Author):

In this work, Uramoto et al. do demonstrate that there is a remarkable abundance of Mn microparticles that have been deposited and preserved in the massive amount of sediments that have slowly accumulated in the highly oligotrophic South Pacific Gyre. This is the key observation in the manuscript. At the large scale, the authors then try to extrapolate their data to other oxic sediments to create a sense of scale for the potential amount of subsurface Mn that is not currently well accounted for in global budget. At the small scale (i.e. scale of particles), the authors show that there is some organic carbon associated with the Fe/Mn particles, and they also try to examine the potential mineralogy and formation mechanism.

It is truly surprising that manganese microparticles are preserved up to 100 million years in subseafloor sediments. The extent to which these particles are primary vs. secondary (i.e. have been modified by diagenetic alteration) is not always clear. At times the authors refer to the microparticles as Fe/Mn particles, presumably mixtures of Fe and Mn oxides that are in those tangles with the organic carbon. It is inferred that these particles form from oxidation and deposition of Fe(II) and Mn(II) dispersed from hydrothermal systems in the deep ocean.

We appreciate Reviewer 3 for providing positive comments and useful suggestions, which significantly improved the manuscript.

Comment #3.1

Yet papers on this topic, which also include x-ray spectromicroscopy characterization of such phases, such as works by Toner et al. on the formation and preservation of carbon-associated Fe-particles and their long distance transport from hydrothermal sites, or such as by Templeton et al. in the formation of Fe/Mn oxides and carbon that get deposited onto rock and sediment surfaces in the deep ocean away from direct hydrothermal input, are not referenced. Some of the microparticle images in Figure 1 in this work even look v. similar to SEM and FIB/TEM images of Fe/Mn-organic phases embedded into basalt surfaces observed in Templeton et al., (2009). Thus it seems that this work should placed within some of the processes examined by those authors.

The authors thank this comment. We cited Templeton et al. (2009), and compared morphological and spectroscopic characteristics of minerals described in the paper to Mn-microparticles of our study. We added the discussion at lines 121-123 in the revised manuscript.

Comment #3.2

In this work, the authors have made efforts to specifically characterize the Mn and carbon speciation. The Mn appears to be MnCO₃, which is likely not primary but rather a secondary product (i.e. from Mn(IV)-reduction). This is mentioned briefly but not necessarily clearly articulated for the reader. Also, is there spectroscopic data of some of the particles that shows Mn(IV), for example in the Fe/Mn rich materials? It would have been very helpful to have bulk Mn XANES for many samples from this study in order to get at average Mn-oxidation state. Instead, we only have very limited data for a few particles that were subjected to Mn L-edge NEXAFS analysis during STXM imaging.

The authors thank to this important comment. We conducted XAFS analysis of the bulk samples and confirmed that manganese in bulk sediments consists mainly of Mn(IV) (Fig. 4). Moreover, we calculated

thermodynamic properties and re-examined mineral characters. Based on the multiple analyses, we concluded that average Mn-oxidation state in Mn-microparticles is Mn(IV). In addition, we conducted the repeated measurement of Mn(IV) reference mineral (δ -MnO₂) and found the changes in redox state of Mn(IV) during the spectroscopic measurement. Taken together, we concluded that Mn-oxidation state in Mn-microparticles is Mn (IV), and Mn (II) signals on NEXAFS/STXM should be from experimental photoreduction during STXM measurement. This means, in the current state of STXM analyses, precise determination of Mn redox state is technically difficult, and we carefully discussed these data interpretations (including the possibility of artificial photoreduction of manganese) in the Discussion part of the revised manuscript (lines 144-167).

Comment #3.3

For the Fe L-edge, why do the authors not try to determine Fe oxidation state? Are these particles primarily Fe(II), Fe(III) or a mixture? A few model compounds are shown, but not enough for a reader to draw a conclusion. There is some excellent data about Fe(II/III) quantification from Fe L-edge XANES in Bourdelle et al. (2013). I expect some of those Toner papers also contain good comparisons.

According to this comment and comment #3.4 below, we conducted additional measurement of Mn-microparticles by STXM and we added explanation for the iron spectra of STXM data of Mn-microparticles (Fig. 5; lines 110-112)

Comment #3.4

Another question about the spectroscopic data. It seems that there are Mn NEXAFS for ABCDEF (three spots, two different particles), and C XANES for ABC (3 spots on just one particle). Given the scope of the samples assessed in this study, why isn't there a much larger data set? It seems like many more particles should have been imaged, giving rise to many more Mn and C spectra for statistical analysis and an assessment of variability in forms of Mn and C that are present.

According to this comment, we conducted additional STXM measurements and described on the stability or variability of each element spectra at lines 105-112, and 144-167 in the revised manuscript.

Comment #3.5

Although the Mn NEXAFS have very good signal to noise and are clearly comparable to the model shown (Mn-carbonate), the C NEXAFS data is quite noisy and hard to compare against models. It would be good if the authors at least pointed out major transitions in the data and showed more clearly how those transitions fit against the model compounds (they invoke humics and fulvics as the best fit). It is very common in such C NEXAFS analysis to identify the actual bonds probed and to point out the well-calibrated energy position of aliphatics, amides, carbonates, etc. Examples can be found in papers by Clara Chan, Julie Cosmidis and many others.

Thanks to pointing out this. Noisy data was due to sample drift during STXM measurement and we made correction on STXM spectra. However, TEM-EDS spectra (Fig. 2) confirmed that C in Mn-microparticles was minor component and corrected data were still hard to compare against models. Consequently, we omitted C spectra from Fig. 5.

Comment #3.6

For the electron diffraction that is presented, is the data being put into a useful framework for proving this is a FeMn mineral? Or are the authors only using the EDS data? Please clarify.

As you mentioned, electron diffraction pattern was used only for phase identification of Mn-oxide (vernadite) as a major mineral component of Mn-particles. A similar spatial distribution of Fe and Mn in X-ray maps suggests that Fe is also contained in the Mn-oxide. We modified the sentences on the TEM-EDS results for better clarity (lines 67-73).

Comment #3.7

The authors do possess measured porewater alkalinity, DIC and Mn profiles, so it would be useful if they would also provide calculations of where MnCO₃ may be supersaturated at various depths.

Thanks for this comment. We calculated thermodynamic properties of the studied oxic sediments, which show that MnCO₃ is unsaturated at various depths, but Mn(IV) minerals is saturated. We added the statement at lines 113-117 in the revised text.

Comment #3.8

The authors mention that there are corresponding microbial data sets. Have they looked into this data to derive any sense if Mn-oxidizers (or reducers) may be present in the sediment as well?

There are interesting data sets showing biologically-driven manganese cycle inside the manganese nodules from northeastern equatorial pacific (Blöthe et al. 2015). Also, Tully and Heidelberg (2013) found that manganese nodules at SPG also harbored similar microbial community. However, both studies detected Mn-specific microbial community only in manganese nodules that is distinct from surrounding sediments. Also, microbial community structure in deeper sediment have not been examined because of very low biomass present in the sediments (D'Hondt et al. 2015). Although it is very interesting, the involvement of microbial activity in diagenetic dissolution of Mn-microparticles is too speculative and we deleted the corresponding sentence in the revised manuscript.

Comment #3.9

There is a huge growth in studies right now that are focused on how Fe and Mn oxides can stabilize organic matter and lead to its preservation, mostly in the soil science vs. oceanography literature. Since the authors are surprised by this phenomenon in their data set, due to the oligotrophic conditions of the South Pacific Gyre, it would be good for them to research this further.

We agree. We cited the soil literature that described organic matter stabilization in Fe and Mn minerals (ref. 4).

Comment #3.10

The manuscript would also benefit from a conceptual model. It would be good to create more of a timeline regarding processes the authors invoke: loading of Fe/Mn-organic tangles in to these sediments, with the silicate fraction, followed by in-situ transformations that occur (that should be specifically defined), and fate of the microparticles with time and the Fe, Mn and C components they contain (e.g. formation of MnCO₃ as one

example). Right now there is too much ambiguity and many readers could misinterpret this study.

The authors thank for this comment. We made a new conceptual figure that summarizes formation and diagenesis of Mn-microparticles in the deep-sea environment (Supplementary Fig. 10).

Reviewers' comments:

Reviewer #1 (Remarks to the Author):

Review of revised version of Uramoto et al.

The manuscript submitted by Uramoto et al. is a revised version of a manuscript that I have evaluated before.

This revised version significantly differs from the initial submission. The manuscript is now much longer than the initial version, has a different structure (text now separated into Results and Discussion) and an additional coauthor (R. Nakada). However, what is most important is that some of the results and findings reported in the initial submission were obviously erroneous or had some serious flaws. In this revised version, the authors have thus put a completely different emphasis after having re-done some of the analyses and performed some additional ones. Based on these new measurements they found that in contrast to their key statement in the initial version that "analyses of STXM showed the predominance of Mn(II) in Mn-microparticles" they have now found mostly Mn(IV).

While the quantification of the amount of Mn-microparticles in the sediments of the South Pacific Gyre as done in this study is definitely an important finding and contributes to better constrain marine element budgets, any sound explanation for the origin or potential formation pathway of the Mn micronodules is missing. At least I did not find one - neither in the manuscript text nor in the abstract. So, how do the authors explain the high abundance of these microparticles in these oxic sediments and why are they only found below the CCD?

The whole discussion is imprecise and confusing and in parts gives the impression that the authors have not fully understood the different formation processes and pathways of ferromanganese minerals in deep sea depositional environments. This is even more surprising since a vast amount of literature on this topic is available since the 1980s and earlier. In addition, there are still numerous typos and imprecise statements and also the English still needs some significant polishing.

Based on the above considerations I therefore regret to say that the manuscript is not suitable for publication in Nature Communications. I have listed my specific comments below and hope that they may be helpful for the authors to revise the manuscript.

Detailed and specific comments

I would suggest to rephrase the first sentence of the abstract as follows:

Ferromanganese minerals are widely distributed at the seafloor of oceanic abyssal plains and in subseafloor sediments. Assessing the input, formation and diagenetic alteration of these minerals is important for understanding the global marine cycles of manganese and associated trace elements.

(I would still use "ferromanganese" minerals here as in the initial version of the manuscript – and not manganese minerals)

Line 35: ... "are found" instead of preserved ...; ... of the "South Pacific Gyre" (SPG) ...

L. 39 "precipitating" instead of precipitated; do you mean "oceanic bottom water"? (instead of deep seawater)? If yes, please say so.

L. 45: I do not fully understand and it is not clear to me what you mean with "critical" in this context?! I also find the wording "environmental material" a bit strange. Please be more precise here.

I also think that in the context of better understanding the marine manganese cycle and budget it is not only manganese itself but also the important role this element plays for the cycling of numerous trace elements. This should be mentioned as well.

L. 76: I would rather speak of "content" here than of "budget".

L. 80: Besides the formation/preservation I would also talk about the "sources" or "origin".

L. 110: So far in this manuscript, the authors have only talked about oxic sediments. Now they state that they have performed analyses both in oxic and anoxic sediments. What kind of sediments are these, from which locations do they come from? I find this very confusing as it stands.

L. 128: Here you state that the elemental composition of the Mn-microparticles is similar to ferromanganese minerals of hydrogenous origin. Please give the respective references. Perhaps it is also better to discuss this in the Discussion chapter?!

Ls. 128 ff.: I do not understand this sentence. "higher magnitude of decrease in the cerium anomaly ..."? Please, rephrase.

Ls. 134 ff: The results reported here contradict the findings reported in the initial version of the manuscript. While in the previous version it was stated that STXM showed the predominance of Mn(II) in Mn-microparticles – it is now described that Mn in the bulk sediment samples (and in the Mn-microparticles as explained in the response to reviewer #2) consists mostly of Mn(IV).

Ls. 137 ff.: I assume from what is stated here, that there have been some severe problems with the STXM method to determine the redox state of Mn in the Mn-microparticles. Also in the response to reviewer #2 the authors explain that there were issues (photoreduction) with the Mn(IV) reference material, which was obviously reduced to Mn(II) during the measurement. So, the main findings of the previous version were not correct.

Ls. 145 ff.: The authors have now also performed thermodynamic calculations based on available pore-water data. The fact that the pore waters are supersaturated with respect to Mn(IV) minerals does not tell you that all the Mn minerals present in the sediments are also in this oxidation state. In contrast, many studies have shown that minerals can be present within the sediments, which are unstable/metastable under the ambient geochemical conditions of the pore water. Moreover, thermodynamic calculations always assume equilibrium conditions – which does not need to be the case. In other words, thermodynamic calculations cannot be used to infer or prove the mineralogy of the real/actual mineral assemblage within a sediment core.

Ls. 156 ff.: Why do the authors find it surprising that Mn is highly concentrated in Mn-microparticles within oxic sediments? This is what I would have expected.

Ls. 160 ff.: Are the authors speaking about hydrogenetic growth here? If yes, please say so. What exactly is meant by "deep seawater"? Do they mean bottom water? If yes, please say so.

Ls. 164 ff.: What kind of redox changes in bottom water are you referring to here? Please, be more precise. Moreover, the reference as given here is not correct. To my knowledge the cited paper does not talk about processes in bottom water but in the sediments/pore waters.

Ls. 166 ff.: I do not understand this sentence at all. It is highly contradictory.

Ls. 173 ff.: In this sentence the link to hydrothermal plumes is not clear to me. The argumentation in this sentence also does not convincingly explain why these Mn-microparticles are only found below the CCD.

Ls. 179 ff.: This sentence is confusing and needs to be clarified.

L. 183: What exactly do you mean with "component mineral" here? This is not clear.

Ls. 182 ff.: As already stated above, the saturation state of the pore water with respect to a specific mineral does not give you proof for the mineral assemblage present in the sedimentary solid phase. It only tells you which mineral may be or is precipitating.

Ls. 185 ff. and 199 ff.: Here the authors clearly state that during the STXM measurements the samples and standard materials are heavily affected by reduction effects – in this way reducing the initially present Mn(IV) to Mn(II). This is also why in the previous version of their manuscript one of their main conclusions was that Mn in the microparticles is mostly present as Mn(II) – which is the opposite to what is now said in this revised version. Why do you use and refer to this method at all if the results produced are not correct – i.e. heavily affected by reduction effects during analyses. This makes no sense to me at all and I would therefore suggest to delete all issues related to STMX from the manuscript.

Ls. 194.: Also the discussion of potential further artifacts during sample preparation is partly odd because – if the Mn is mostly present in the form of Mn(IV) in situ – none of the effects mentioned (which would be oxidation) should affect the already fully oxidized Mn.

Ls. 203 ff.: Here they mention the "formation process". However, so far in the manuscript I have not read any convincing formation pathway of the observed Mn-microparticles.

Ls. 207-208.: You are talking about the "total manganese in the bulk sediment" here. If the Mn contained in the Mn-microparticles decreases with depth – in which form is the rest (Mn not contained in the microparticles) present in the sediment – in particular in the deeper parts of the sediments?

Ls. 208 ff.: Did you really detect dissolved Mn in pore water in these oxic sediments? Moreover, I find the way the pore water data are presented in Fig. 8a of the supplement extremely confusing and cannot really see a general increase in concentrations at all sites. It is really hard to tell if the profiles of O₂ and Mn²⁺ are not plotted versus depth.

L. 213: How can Mn(IV) be reduced if the sediments are fully oxic? – as you say throughout the manuscript? Please, explain.

Ls. 215 ff.: And what does a decrease in the Ce anomaly tell you? This is not clear. Please discuss and support by relevant references.

L. 219: Again: in which form is the rest of the Mn(IV) present in the sediment?

L. 219 ff.: Why are only the Mn-microparticles redox-sensitive? This makes no sense. What about the rest of the Mn(IV) contained in the bulk sediment? And why should Mn be reduced anyhow if the investigated sediments are all oxic throughout as you say? Please explain!

Ls. 233 ff.: As already mentioned above, I have not seen any clear and convincing explanation for the formation/preservation of these Mn-microparticles in this manuscript.

Acknowledgements: Please, also thank your reviewers.

I have not corrected the English – but there are still many flaws and the manuscript needs careful polishing and/or check by an English native speaker.

Reviewer #2 (Remarks to the Author):

The authors have satisfactorily responded to all my initial concerns with the manuscript. The final product is a highly improved manuscript, of which the results should be of great interest to the oceanographic science community. My only minor concern is that there are still a number of grammatical corrections that are necessary prior to publication, particularly in the revised sections of the manuscript. These are all minor corrections that are needed but would improve the readability of the article.

Reviewer #3 (Remarks to the Author):

In this work, Uramoto et al. do still demonstrate that there is a remarkable abundance of Mn microparticles in oxic deep sea sediments, thereby changing the structure/balance of the global manganese budget.

The authors have addressed numerous questions and suggestions from the reviewers. In terms of data presentation and interpretation, I specifically note that this team has added more spectroscopic analyses to confirm the speciation of manganese. Indeed, the majority of manganese is in the Mn(IV) form, as convincingly determined through bulk XAS analyses. I do have two follow-up questions about the data as it is now presented: why is STXM data shown in primary figures vs. supplementary figures? I don't strongly object, but I find it curious now that the authors have proven to themselves that STXM on the Mn-oxide particles causes beam reduction and the false detection of Mn(II)-minerals? It seems that this should be noted as an important experimental observation but that the key data for the reader is the true Mn speciation as revealed by bulk Mn XAS? The second related question is how confident are the authors that the Mn(IV)-phase is vernadite specifically? When reporting the spectroscopic data, they note that the Mn XANES are similar to δ -MnO₂ and/or birnessite. From electron diffraction, the phases are poorly crystalline. From EDS, there is Fe with Mn. So is the Fe incorporation the key criteria and is it sufficiently robust to provide a mineral identification of "vernadite" vs. a more general description of a Mn (or ferromanganese) oxide? A little further explanation for reader would likely be helpful given how important this Mn-oxide phase is to the whole story.

Brief note: please define SPG (South Pacific Gyre) in abstract

Response to the reviewers' comments:

Reviewer #1:

The manuscript submitted by Uramoto et al. is a revised version of a manuscript that I have evaluated before.

This revised version significantly differs from the initial submission. The manuscript is now much longer than the initial version, has a different structure (text now separated into Results and Discussion) and an additional coauthor (R. Nakada). However, what is most important is that some of the results and findings reported in the initial submission were obviously erroneous or had some serious flaws. In this revised version, the authors have thus put a completely different emphasis after having re-done some of the analyses and performed some additional ones. Based on these new measurements they found that in contrast to their key statement in the initial version that “analyses of STXM showed the predominance of Mn(II) in Mn-microparticles” they have now found mostly Mn(IV).

While the quantification of the amount of Mn-microparticles in the sediments of the South Pacific Gyre as done in this study is definitely an important finding and contributes to better constrain marine element budgets, any sound explanation for the origin or potential formation pathway of the Mn micronodules is missing. At least I did not find one - neither in the manuscript text nor in the abstract. So, how do the authors explain the high abundance of these microparticles in these oxic sediments and why are they only found below the CCD?

The whole discussion is imprecise and confusing and in parts gives the impression that the authors have not fully understood the different formation processes and pathways of ferromanganese minerals in deep sea depositional environments. This is even more surprising since a vast amount of literature on this topic is available since the 1980s and earlier. In addition, there are still numerous typos and imprecise statements and also the English still needs some significant polishing.

Based on the above considerations I therefore regret to say that the manuscript is not suitable for publication in Nature Communications. I have listed my specific comments below and hope that they may be helpful for the authors to revise the manuscript.

We wish to sincerely thank Reviewer 1 for providing insightful comments, which significantly improved the manuscript. In the revised version of the manuscript, we have addressed all of your comments and suggestions; in particular, we have revised our discussion concerning the multiple Mn-microparticles formation/preservation processes.

Detailed and specific comments

Comment #1.1

I would suggest to rephrase the first sentence of the abstract as follows:

Ferromanganese minerals are widely distributed at the seafloor of oceanic abyssal plains and in subseafloor sediments. Assessing the input, formation and diagenetic alteration of these minerals is important for understanding the global marine cycles of manganese and associated trace elements.

(I would still use “ferromanganese” minerals here as in the initial version of the manuscript – and not manganese minerals)

According to your suggestion, we rephrased the **Abstract** (lines 22-25). We also rephrased “manganese minerals” to “ferromanganese minerals” throughout the manuscript.

Comment #1.2

Line 35: ... “are found” instead of preserved ...; ... of the “South Pacific Gyre” (SPG) ...

Thank you for your feedback; both have been corrected (line 27-28).

Comment #1.3

L. 39 “precipitating” instead of precipitated; do you mean “oceanic bottom water”? (instead of deep seawater)? If yes, please say so.

This has been corrected (line 31); it did refer to oceanic bottom water. We rephrased the wording (line 32).

Comment #1.4

L. 45: I do not fully understand and it is not clear to me what you mean with “critical” in this context?! I also find the wording “environmental material” a bit strange. Please be more precise here.

I also think that in the context of better understanding the marine manganese cycle and budget it is not only manganese itself but also the important role this element plays for the cycling of numerous trace elements. This should be mentioned as well.

Thank you. To address these points, we rephrased the sentences as follows: “Ferromanganese mineral deposits are extremely abundant in marine and terrestrial environments; thus, understanding the redox-sensitive dynamics and budget of ferromanganese minerals are important for understanding the global cycling of Mn and numerous associated trace elements” (lines 39-41).

Comment #1.5

L. 76: I would rather speak of “content” here than of “budget”.

Thank you for your feedback; this has been corrected (line 72).

Comment #1.6

L. 80: Besides the formation/preservation I would also talk about the “sources” or “origin”.

Thank you for your helpful comments. Your comments (together with comments #1.8, #1.9, #1.17, #1.24, #1.25, #1.26, #1.27, and #1.30) helped us to develop more detailed discussion of the sources and origins of the manganese in Mn-microparticles. We revised our discussions of the formation/preservation of Mn-microparticles based on further examination of multiple manganese mineral formation, source, and preservation processes in entirely oxic sediments.

Comment #1.7

L. 110: So far in this manuscript, the authors have only talked about oxic sediments. Now they state that they have performed analyses both in oxic and anoxic sediments. What kind of sediments are these, from which locations do they come from? I find this very confusing as it stands.

Thank you for your feedback. To address this point, we have changed the text to stress that this study analyzed not only oxic sediments, but also anoxic sediments as references in the **Introduction** (lines 64-67).

Comment #1.8

L. 128: Here you state that the elemental composition of the Mn-microparticles is similar to ferromanganese minerals of hydrogenous origin. Please give the respective references. Perhaps it is also better to discuss this in the Discussion chapter?!

Per your comment, we revised the Mn-microparticle composition based on the representative references in the **Results** (lines 128-130, Fig. 4e). We also revised the discussion of the formation of Mn-microparticles based on multiple processes, and specifically, changes in the degree of hydrogenetic and hydrothermal processes during the tectonic migration of the study sites away from the oceanic ridge (lines 167-205).

Comment #1.9

Ls. 128 ff.: I do not understand this sentence. “higher magnitude of decrease in the cerium anomaly ...”? Please, rephrase.

In a previous manuscript, we described this change in the cerium anomaly as an indicator of diagenesis in Mn-microparticles. However, during the course of revision according to your helpful comments, we decided that discussion of the decrease in the cerium anomaly and diagenetic reduction of Mn(IV) oxides used in the previous manuscript was not appropriate. Based on a revised examination of composition characteristics (Fig. 4e), we amended the text to state that Mn-microparticles were not affected by diagenesis in the fully oxic sediments (lines 237-238). We propose that the Mn-microparticle cerium anomaly represents a gradual change in the Mn-microparticle formation mechanism from more hydrothermal to more hydrogenetic

during the tectonic migration of the study sites away from the oceanic ridge (e.g., East Pacific Rise) (lines 167-205).

Comment #1.10

Ls. 134 ff: The results reported here contradict the findings reported in the initial version of the manuscript. While in the previous version it was stated that STXM showed the predominance of Mn(II) in Mn-microparticles – it is now described that Mn in the bulk sediment samples (and in the Mn-microparticles as explained in the response to reviewer #2) consists mostly of Mn(IV).

In response to the initial review comments by Reviewers 2 and 3, we conducted XAFS analysis of the bulk sediments and porewater thermodynamic calculations. Also, we re-examined the TEM-EDS, electron diffraction, and ICP-MS results. These additional and re-examined data showed that the chemical state of manganese in Mn-microparticles was Mn(IV); we revised the manuscript accordingly. In addition, we repeated the analysis of the standard Mn(IV) oxide samples, the results of which suggested photoreduction during the manganese STXM measurements (Supplementary Figure 11). Based on this multidisciplinary examination, we concluded that the manganese STXM results indicate photoreduction during sample analysis in the 1st revision of the manuscript. In this 2nd revision of the manuscript, we moved these data to the Supplement as Reviewer 3 suggested, as the STXM results are now not key for the discussion, but can be of assistance to the readers in explaining the unintended photoreduction via STXM.

Comment #1.11

Ls. 137 ff.: I assume from what is stated here, that there have been some severe problems with the STXM method to determine the redox state of Mn in the Mn-microparticles. Also in the response to reviewer 2, the authors explain that there were issues (photoreduction) with the Mn(IV) reference material, which was obviously reduced to Mn(II) during the measurement. So, the main findings of the previous version were not correct.

As described in our response to comment #1.10, STXM results for manganese in Mn-microparticles indicated artificial photoreduction, and these results are not key to the major conclusions drawn herein. However, as suggested by Reviewer 3, these data are important for supporting the analytical rigor herein; STXM results have been moved to Supplementary Figure 11 (please see our response to #1.10)

Comment #1.12

Ls. 145 ff.: The authors have now also performed thermodynamic calculations based on available pore-water data. The fact that the pore waters are supersaturated with respect to Mn(IV) minerals does not tell you that all the Mn minerals present in the sediments are also in this oxidation state. In contrast, many studies have shown that minerals can be present within the sediments, which are unstable/metastable under

the ambient geochemical conditions of the pore water. Moreover, thermodynamic calculations always assume equilibrium conditions – which does not need to be the case. In other words, thermodynamic calculations cannot be used to infer or proof the mineralogy of the real/actual mineral assemblage within a sediment core.

Thank you again for your valuable comments. Based on this comment, we reconsidered the state of the Mn-microparticles in the sediments. In our 2nd revision of the manuscript, we used thermodynamic calculations to discuss the formation potential of Mn(II) minerals in the sediment, and included these results in the discussion of the STXM results (lines 219-236). The revised manuscript now states in the **Results** (lines 151-156) that the thermodynamic calculations can be used to indicate the state of saturation in the porewater.

Comment #1.13

Ls. 156 ff.: Why do the authors find it surprising that Mn is highly concentrated in Mn-microparticles within oxic sediments? This is what I would have expected.

Thank you; we agree that “surprising” was an overstatement. We rephrased the sentence to describe the important finding of this study: specifically, that manganese was concentrated in Mn-microparticles, which has implications for the manganese budget in the subseafloor environment (lines 162-166).

Comment #1.14

Ls. 160 ff.: Are the authors speaking about hydrogenetic growth here? If yes, please say so.

What exactly is meant by “deep seawater”? Do they mean bottom water? If yes, please say so.

Thank you for your feedback. We meant to discuss hydrogenetic growth in bottom seawater. We rephrased the wording (line 167-168).

Comment #1.15

Ls. 164 ff.: What kind of redox changes in bottom water are you referring to here? Please, be more precise. Moreover, the reference as given here is not correct. To my knowledge the cited paper does not talk about processes in bottom water but in the sediments/pore waters.

In the course of revision based on your valuable comments, we reached the conclusion that redox changes in water mass were not related to the formation of Mn-microparticles; thus, we deleted the corresponding discussion of the redox state of bottom water in the revised manuscript.

Comment #1.16

Ls. 166 ff.: I do not understand this sentence at all. It is highly contradictory.

Thank you again for your useful comments. In our revision, we reconsidered the source, origin, and preservation of Mn-microparticles in bottom seawater and sediment; our revised mechanism

does not preclude formation of Mn-microparticles in the photic zone or minimum-oxygen zone. We have deleted this part of the discussion in the revised manuscript.

Comment #1.17

Ls. 173 ff.: In this sentence the link to hydrothermal plumes is not clear to me. The argumentation in this sentence also does not convincingly explain why these Mn-microparticles are only found below the CCD.

After reconsidering the sources, origins, and preservation of Mn-microparticles in bottom seawater and sediment per your valuable comments and suggestion, we concluded that the compositional characteristics in the revised Fe-Mn-(Cu+Ni+Co)*10 diagram and the negative cerium anomaly in several Mn-microparticle samples indicated a hydrothermal effect on the formation of some Mn-microparticles (lines 191-202).

Furthermore, our revised Mn-microparticle formation mechanism now does not preclude formation of such particles above the CCD. In the revised manuscript, we attributed the apparent absence of Mn-microparticles to two possible mechanisms: (1) a higher calcareous ooze sedimentation rate, which may have produced concentrations below the detection limit of the SEM-based observation herein, and (2) possible adsorption of dissolved manganese in bottom water into carbonate-rich sediments above the CCD that hampered the formation of Mn-microparticles (lines 210-218).

Comment #1.18

Ls. 179 ff.: This sentence is confusing and needs to be clarified.

According to your comment, we rephrased the sentence as follows “Bulk sample XAFS analysis indicates the manganese oxidative state, while separate Mn-microparticle STXM analysis indicates the manganese reductive state.” (lines 220-222).

Comment #1.19

L. 183: What exactly do you mean with “component mineral” here? This is not clear.

We meant vernadite, as discussed in a previous manuscript. However, as also suggested by Reviewer 3, our results clearly indicated only a poorly crystalline mineral phase, and the identification itself was not robust. Therefore, we rephrased “vernadite” to “ferromanganese oxide” throughout the manuscript and we rephrased this sentence as follow: “The Mn-microparticles consist primarily of ferromanganese oxides” (lines 222-223).

Comment #1.20

Ls. 182 ff.: As already stated above, the saturation state of the pore water with respect to a specific mineral does not give you proof for the mineral assemblage present in the sedimentary solid phase. It only tells you which mineral may be or is precipitating.

Thank you again for your valuable comments on the thermodynamic calculations. We rephrased the sentence per your suggestion (lines 151-156) (also please see our response to #1.12)

Comment #1.21

Ls. 185 ff. and 199 ff.: Here the authors clearly state that during the STXM measurements the samples and standard materials are heavily affected by reduction effects – in this way reducing the initially present Mn(IV) to Mn(II). This is also why in the previous version of their manuscript one of their main conclusions was that Mn in the microparticles is mostly present as Mn(II) – which is the opposite to what is now said in this revised version. Why do you use and refer to this method at all if the results produced are not correct – i.e. heavily affected by reduction effects during analyses. This makes no sense to me at all and I would therefore suggest to delete all issues related to STMX from the manuscript.

Thank you for this useful suggestion. Because, to our knowledge, previous studies have not shown such heavy photoreduction during manganese STXM analysis, we propose that the STXM manganese results should be included for the edification of the reader, as such techniques may be used in future analyses. As Reviewer 3 suggested, the STXM results have been moved to Supplementary Figure 11 (please also see our response to #1.10).

Comment #1.22

Ls. 194.: Also the discussion of potential further artifacts during sample preparation is partly odd because – if the Mn is mostly present in the form of Mn(IV) in situ – none of the effects mentioned (which would be oxidation) should affect the already fully oxidized Mn.

Thank you for this feedback. In the course of our revision, we concluded that oxidation artifacts were not important for the redox state of the manganese in the samples herein; thus, we deleted the corresponding discussion.

Comment #1.23

Ls. 203 ff.: Here they mention the “formation process”. However, so far in the manuscript I have not read any convincing formation pathway of the observed Mn-microparticles.

In the course of revision based on your comments (together with comments #1.8, #1.9, #1.17, #1.24, #1.25, #1.26, #1.27, and #1.30), we revised all discussion of Mn-microparticle formation based on the changes in the degree of hydrogenetic and hydrothermal processing outlined in the **Discussion** (lines 167–205).

Comment #1.24

Ls. 207-208.: You are talking about the “total manganese in the bulk sediment” here. If the Mn contained in the Mn-microparticles decreases with depth – in which form is the rest (Mn not contained in the microparticles) present in the sediment – in particular in the deeper parts of the sediments?

According to this comment, we examined the manganese-concentrated minerals not contained in Mn-microparticles. As shown in Fig. 2a-b and Supplementary Fig. 9, the rest of the manganese minerals observed in this study consisted primarily of irregular-shaped minerals. Such minerals can be observed all analyzed samples; thus, we conclude that both Mn-microparticles and the rest of the Mn(IV) minerals contribute to the manganese budget in subseafloor environments. However, these irregular-shaped minerals show size variations that are not compatible with our flow cytometry-based selective particle separation technique. Therefore, we focused on the analysis of recoverable manganese minerals in this study. We have added a statement of this in the **Results** (Lines 118-127).

Comment #1.25

Ls. 208 ff.: Did you really detect dissolved Mn in pore water in these oxic sediments? Moreover, I find the way the pore water data are presented in Fig. 8a of the supplement extremely confusing and cannot really see a general increase in concentrations at all sites. It is really hard to tell if the profiles of O₂ and Mn²⁺ are not plotted versus depth.

The data on dissolved Mn in pore water was taken from a previous publication that measured porewater extracted from oxic sediments during IODP Expedition 329. In our revised manuscript, we concluded, based on your valuable comments, that the Mn-microparticles were stable in the sediment, which makes the dissolved Mn irrelevant to our discussion of the state of the Mn-microparticles in the sediment. We therefore deleted Fig. 8a.

Comment #1.26

L. 213: How can Mn(IV) be reduced if the sediments are fully oxic? – as you say throughout the manuscript? Please, explain.

A reexamination of the composition of the Mn-microparticles (Fig. 4e), along with your critical comments (#1.8, #1.9, and #1.25), led us to conclude that the discussion on the reduction of Mn(IV) was not appropriate; thus, we deleted the corresponding discussion.

Comment #1.27

Ls. 215 ff.: And what does a decrease in the Ce anomaly tell you? This is not clear. Please discuss and support by relevant references.

Our revised Mn-microparticle composition characteristics along with your comments (#1.8, #1.9, #1.25, and #1.26), indicated that the Ce anomaly indicates formation processes of Mn-microparticles, not diagenesis. Therefore, we deleted the discussion of diagenetic changes in the Ce anomaly. In the revised manuscript, we suggest that the Ce anomaly provides insight into changes in hydrogenesis and hydrothermal effects on the formation of Mn-microparticles, based on additional references (lines 171-174, 194-198).

Comment #1.28

L. 219: Again: in which form is the rest of the Mn(IV) present in the sediment?

According to your comment (and #1.24), we examined the rest of the manganese minerals in sediments. As shown in Fig. 2a-b and Supplementary Fig. 9, the rest of the manganese minerals observed in this study consisted primarily of irregular-shaped minerals; such minerals were observed in all the analyzed samples and indicate that both Mn-microparticles and the rest of Mn(IV) minerals contribute to the manganese budget in seafloor environments. However, these irregular-shaped minerals show size variations that are not compatible with our flow cytometry-based selective particle separation technique. Therefore, we focused on the analysis of recoverable manganese minerals in this study. We have added a statement of this in the **Results** (Lines 118-127) (also please see our response to #1.24)

Comment #1.29

L. 219 ff.: Why are only the Mn-microparticles redox-sensitive? This makes no sense. What about the rest of the Mn(IV) contained in the bulk sediment? And why should Mn be reduced anyhow if the investigated sediments are all oxic throughout as you say? Please explain!

Thank you. As you mention, both Mn-microparticle and the rest of Mn(IV) oxides are redox-sensitive. We deleted the corresponding wording in the previous version of the manuscript. Furthermore, per with your valuable comments (#1.8, #1.9, #1.25, and #1.26), we deleted all discussion of the reduction of Mn-microparticles, based on a revised examination of Mn-microparticle composition characteristics in the revised ternary diagram (Fig. 4e), which clearly show that Mn-microparticles in the fully oxic sediments were not affected by diagenesis.

Comment #1.30

Ls. 233 ff.: As already mentioned above, I have not seen any clear and convincing explanation for the formation/preservation of these Mn-microparticles in this manuscript.

Thank you for your very important and encouraging comments. Your comments stimulated a revised discussion of Mn-microparticle formation and preservation processes. We revised the figures and discussion related to Mn-microparticle formation and preservation processes herein with reference to comments #1.6, #1.8, #1.9, #1.17, #1.24, #1.25, #1.26, and #1.27 (lines 167-218).

Regarding formation, we concluded that Mn-microparticles were mainly formed by hydrogenetic precipitation mixed with hydrogenetic-hydrothermal processes occurring in old sediment formed when the site was near the oceanic ridge (lines 191-198). We thus infer that changes in the formative processes of Mn-microparticles were caused by the tectonic migration of the study sites away from the oceanic ridge.

Regarding preservation, we concluded that Mn-microparticles in fully oxic sediments were not affected by diagenesis (Fig. 4e). Mn-microparticles were stabilized in oxic sediments after deposition. We discuss the input of Mn-microparticles further at Lines 240-248.

Comment #1.31

Acknowledgements: Please, also thank your reviewers.

Thank you again for your comments! According to your comment, we have acknowledged the reviewers for their insightful review (lines 419-420).

Comment #1.32

I have not corrected the English – but there are still many flaws and the manuscript needs careful polishing and/or check by an English native speaker.

Thank you; per your feedback, a native English speaker has proofread our manuscript.

Reviewer #2:

The authors have satisfactorily responded to all my initial concerns with the manuscript. The final product is a highly improved manuscript, of which the results should be of great interest to the oceanographic science community. My only minor concern is that there are still a number of grammatical corrections that are necessary prior to publication, particularly in the revised sections of the manuscript. These are all minor corrections that are needed but would improve the readability of the article.

Thank you for your comment; per your feedback, a native English speaker has proofread our manuscript.

Reviewer #3:

In this work, Uramoto et al. do still demonstrate that there is a remarkable abundance of Mn microparticles in oxic deep sea sediments, thereby changing the structure/balance of the global manganese budget.

We wish to thank Reviewer 3 for providing positive comments and useful suggestions, both of which improved the manuscript.

Comment #3.1

The authors have addressed numerous questions and suggestions from the reviewers. In terms of data presentation and interpretation, I specifically note that this team has added more spectroscopic analyses to confirm the speciation of manganese. Indeed, the majority of manganese is in the Mn(IV) form, as convincingly determined through bulk XAS analyses. I do have two follow-up questions about the data as it is now presented: why is STXM data shown in primary figures vs. supplementary figures? I don't strongly object, but I find it curious now that the authors have proven to themselves that STXM on the Mn-oxide particles causes beam reduction and the false detection of Mn(II)-minerals? It seems that this should be noted as an important experimental observation but that the key data for the reader is the true Mn speciation as revealed by bulk Mn XAS?

Thank you for your suggestion. We agree that the STXM results are not crucial to the discussion and have thus moved the STXM results to Supplementary Figure 11.

Comment #3.2

The second related question is how confident are the authors that the Mn(IV)-phase is vernadite specifically? When reporting the spectroscopic data, they note that the Mn XANES are similar to d-MnO₂ and/or birnessite. From electron diffraction, the phases are poorly crystalline. From EDS, there is Fe with Mn. So is the Fe incorporation the key criteria and is it sufficiently robust to provide a mineral identification of "vernadite" vs. a more general description of a Mn (or ferromanganese) oxide? A little further explanation for reader would likely be helpful given how important this Mn-oxide phase is to the whole story.

As you mention, a broad electron diffraction ring indicates a poorly crystalline structure. The interplanar spacing of the mineral contained in the Mn-microparticle is similar to vernadite, but these patterns are not sufficiently robust to identify the mineral. Therefore, we changed "vernadite" to the general descriptive term "ferromanganese oxide" throughout the manuscript and added further description of the mineral in the **Results** (lines 93-96). We also deleted the discussion on vernadite formation processes in the **Discussion**.

Comment #3.3

Brief note: please define SPG (South Pacific Gyre) in abstract

Thank you for your useful comments; this has been corrected.

Reviewers' comments:

Reviewer #1 (Remarks to the Author):

I have now evaluated this manuscript for the third time. I still find the data set extremely impressive and think that the finding and quantification of the amount of Mn contained in microparticles in these abyssal oxic sediments is a very important result and will definitely contribute to better constrain the marine Mn budget. However, the discussion is still very imprecise in parts. I had the impression that in every new version of the manuscript the authors present a new explanation for the input/formation of the observed Mn microparticles – this is indication that the „story“ might need some more time to mature.

The manuscript presents novel findings that – without doubt - are of interest for the broader marine geochemistry and oceanography community. However, before publication of this manuscript, there is definitely some more polishing needed. I have listed my detailed comments below and hope that they will be helpful in preparing a revised version.

Minor comments

L. 32: has to be „bottom water“

L. 32: It is not clear from the abstract how you know that the observed Mn microparticles precipitate or better: have precipitated from bottom water.

L. 36: delete „extremely“

L. 43: „manganese nodules“ instead of manganese deposits

L. 47: This is imprecise. Crust do not form on the sediment but at seamounts.

L. 64 ff.: It is not clear to me why you use „anoxic“ sediments as a reference if the sediments you have investigated in this study are entirely oxic – as stated at the beginning of the sentence.

Ls. 87/88: ... high contents of manganese in these microparticles

L. 92: ... the existence of „a“ poorly

L. 102: This sentence is contradictory and misleading. I do not understand why you talk about anoxic sediments here as you have stated in your manuscript so far that the sediments of the SPG are entirely oxic. Please correct this accordingly.

L. 114: I am not entirely sure what you mean here with „manganese percentage“? Wt. % of Mn in the bulk sediment or in the the microparticles? Please specify.

L. 117: I do not understand the last part of this sentence. „none of the internal structure“?

L. 118: „size“ instead of scale

L. 118: I find it a bit confusing that you now speak of „irregular-shaped“ (by the way – I think it has to be irregularly shaped). Do these particles belong tot he group of „tangled fibrous strands of manganese and iron minerals“? This is not clear to me.

L. 120: Why do you explicitly mention „oxic sediments“ here. I thought the sediments you have investigated are entirely oxic?! So, now I see that obbvously you argue that these Mn particles of irregular shapes do not belong tot he Mn-microparticles?!

L. 144: What does the abbreviation „NEXAFS“ precisely stand for?

L. 158/159: I disagree with the last part of this sentence.

L. 167: Has to be „bottom water“.

Ls. 174 ff.: I am sorry but I do not understand the argumentation in this sentence. First, you state that the particles were formed by hydrogenetic precipitation – but in the last part of the sentence you suddenly mention „hydrothermal“ processes? This line of argumentation is not at all clear to me.

Ls. 181/182: I do not understand this part of the sentence. Moreover, the geochemical conditions in the Peru Basin, referred to here are completely different to those of the SPG. And you also say so.

L. 188: Which „hypotheses“ precisely do you mean here? This is not clear to me.

L. 190: What is a „redox-condition transition area“?

Ls. 204/205: Do you really think that all these potential sources are relevant sources of Mn in your

study area?

Ls. 206 ff.: Here you „suddenly“ speak of anoxic sediments. I thought you only investigated oxic sediments?! Moreover, be precise when speaking of anoxic sediments. In most settings, anoxic sediments are found at greater depth while the upper surface of the sediments is characterized by an oxic zone. This zone prevents the diffusive escape of Mn^{2+} from the sediment into the bottom water. Thus, the argumentation in this sentence is imprecise.

Ls. 222/223: What precisely do you mean with „the manganese reductive state“ and the „manganese oxidative state“ in this context? The wording is a bit odd. Isn't it rather „oxidized“ or „reduced“?!

L. 238: „within the sediment“

Ls. 247 ff.: I would suggest to delete these two sentences because you have no evidence for this assumption. I am sure that measurements of the amount of particulate Mn present over water depth exists in several ocean areas. Are your assumptions in line with such measurements?

L. 265: I do not agree that you have really discussed manganese „dynamics over geologic time“. Therefore, I would suggest to delete this last part of the sentence.

Response to the reviewers' comments

Reviewer #1 (Remarks to the Author):

I have now evaluated this manuscript for the third time. I still find the data set extremely impressive and think that the finding and quantification of the amount of Mn contained in microparticles in these abyssal oxic sediments is a very important result and will definitely contribute to better constrain the marine Mn budget. However, the discussion is still very imprecise in parts. I had the impression that in every new version of the manuscript the authors present a new explanation for the input/formation of the observed Mn microparticles – this is indication that the „story“ might need some more time to mature.

The manuscript presents novel findings that – without doubt - are of interest for the broader marine geochemistry and oceanography community. However, before publication of this manuscript, there is definitely some more polishing needed. I have listed my detailed comments below and hope that they will be helpful in preparing a revised version.

We sincerely thank Reviewer #1 for providing insightful comments and suggestions. All the points were extremely valuable and thus incorporated into the revised version here. As the result, we believe the manuscript has been significantly improved. Our responses to the points raised by you are described here:

Minor comments

L. 32: has to be „bottom water“

Corrected (line 32).

L. 32: It is not clear from the abstract how you know that the observed Mn microparticles precipitate or better: have precipitated from bottom water.

Thank you for your feedback. To address this point, we rephrased this part as follows:

“Three-dimensional micro-texture, and major and trace element compositional analyses revealed that these Mn-microparticles consist of poorly crystalline ferromanganese oxides precipitating from bottom water” (lines 30-32).

L. 36: delete „extremely“

Deleted (line 37).

L. 43: „manganese nodules“ instead of manganese deposits

Corrected (line 44).

L. 47: This is imprecise. Crust do not form on the sediment but at seamounts.

Thank you for your feedback; statements on crust were deleted (line 48).

L. 64 ff.: It is not clear to me why you use „anoxic“ sediments as a reference if the sediments you have investigated in this study are entirely oxic – as stated at the beginning of the sentence.

Thank you for your feedback; analyses of anoxic sediments as a “reference” was imprecise, for which we intended to say “as comparison”. To understand global distribution of Mn microparticles, we analyzed and compared marine sediment microstructures in various redox condition including oxic and anoxic sediments. We therefore rephrased corresponding part as follows: “we analyzed and compared marine sediment core samples in various redox condition through continental margin to open ocean gyre (Supplementary Fig. 1) using a resin-embedding technique¹⁷ involving a biological method that retains the fine texture of both mineral grains and organic materials in sediments at the submicrometer-scale.” (lines 63-66). Together with your other comments, we also revised sentences that described and/or discussed on oxic/anoxic sediment microstructures throughout the manuscript (lines 100-102, 117-119, 196-199, and 243-244)

Ls. 87/88: ... high contents of manganese in these microparticles

Corrected (lines 88-89).

L. 92: ... the existence of „a“ poorly
Corrected (line 92).

L. 102: This sentence is contradictory and misleading. I do not understand why you talk about anoxic sediments here as you have stated in your manuscript so far that the sediments of the SPG are entirely oxic. Please correct this accordingly.

To understand global distribution of Mn-microparticles discovered in this study, we analyzed and compared marine sediment microstructures in various redox condition including oxic and anoxic sediments (please also refer to our response to your comment #1.6). And we observed two different types of microparticles in this study; (a) clay-microparticles and (b) Mn-microparticles (this observation has been in the beginning of the result section, lines 83-87 in revised manuscript). (a) were found in both oxic and anoxic sediments and (b) were found only in oxic sediments. To make things clearer, we separated this sentence into two sentences that describe (1) abundance of clay-microparticles in both oxic and anoxic sediments, and (2) abundance of Mn-microparticles in oxic sediments (lines 100-102).

L. 114: I am not entirely sure what you mean here with „manganese percentage“? Wt. % of Mn in the bulk sediment or in the the microparticles? Please specify.

It means “in the Mn-microparticles”. We corrected the sentence (line 114).

L. 117: I do not understand the last part of this sentence. „none of the internal structure“?

In this part, we described about other type of mineral particles with high Mn-content than Mn-microparticles of tangled fibrous strands. It described manganese-concentrated particles without concentric growth structure nor fibrous structure. To make this point clear, we rephrased these mineral particles as “structureless manganese particles”. We rephrased the corresponding wordings in the text (lines 117-118, 120, 123, 124, 127, and 244)

L. 118: „size“ instead of scale
Corrected (line 120).

L. 118: I find it a bit confusing that you now speak of „irregular-shaped“ (by the way – I think it has to be irregularly shaped). Do these particles belong to the group of „tangled fibrous strands of manganese and iron minerals“? This is not clear to me.

As in our response to your comment above, we described about other type of mineral particles with high manganese content than Mn-microparticles of tangled fibrous strands. We added further description as follows: “We also found manganese-concentrated grains without concentric growth structure nor fibrous structure (structureless manganese particles) which were outside of Mn-microparticles with tangled fibrous strands” (lines 117-119). We also revised Fig. 2 to provide enlarged cross-sectional SEM images of resin-embedded samples which show both Mn-microparticles with tangled fibrous strands and structureless manganese particles (Fig. 2a-d).

L. 120: Why do you explicitly mention „oxic sediments“ here. I thought the sediments you have investigated are entirely oxic?! So, now I see that obviously you argue that these Mn particles of irregular shapes do not belong to the Mn-microparticles?!

This is because, here, we described characteristic manganese mineral particles in oxic sediments based on the investigation of both oxic pelagic sediments and anoxic continental margin sediments. As described in our response to your comments above, structureless manganese grains (“irregular-shaped minerals” in previous manuscript) are outside of Mn-microparticles of tangled fibrous strands and we added statement of this (lines 117-119).

L. 144: What does the abbreviation „NEXAFS“ precisely stand for?

We provided the complete expression of NEXAFS accordingly (line 145).

L. 158/159: I disagree with the last part of this sentence.

We deleted this sentence according to your comment (lines 156).

L. 167: Has to be „bottom water“.

Corrected (line 168).

Ls. 174 ff.: I am sorry but I do not understand the argumentation in this sentence. First, you state that the particles were formed by hydrogenetic precipitation – but in the last part of the sentence you suddenly mention „hydrothermal“ processes? This line of argumentation is not at all clear to me.

Thank you for your feedback; discussion on hydrothermal processes in this sentence was not appropriate. We deleted the sentence (line 176).

Ls. 181/182: I do not understand this part of the sentence. Moreover, the geochemical conditions in the Peru Basin, referred to here are completely different to those of the SPG. And you also say so.

Thank you for your feedback; discussion based on the comparison with the Peru Basin was not appropriate here. We deleted the corresponding sentence (line 176), and corresponding description in Results and figure (previous Supplementary Fig. 10a-b).

L. 188: Which „hypotheses“ precisely do you mean here? This is not clear to me.

Thank you for this feedback; wording “hypotheses” was imprecise here. We described about indication of metallic element mobilization in Mn-microparticles based on the examination of compositional characteristics shown in ternary diagram (Fig. 4e). We rephrased the corresponding sentences as follows: “However, it should be noted that hydrogenetic Mn-microparticles show varying metallic element composition (Fig. 4e). This could be due to remobilization of metals in association with the degradation of organic matter in seafloor environments. This indication of metallic element mobilization in Mn-microparticles should be verified in the future through the sampling and analysis of sedimentary Mn-microparticles between continental margin and open ocean gyre” (lines 177-181)

L. 190: What is a „redox-condition transition area“?

It means ocean area between continental margin and open ocean gyre. We rephrased the wordings (line 181).

Ls. 204/205: Do you really think that all these potential sources are relevant sources of Mn in your study area?

After reconsidering the source of manganese of Mn-microparticles in the South Pacific Gyre per your valuable comments and suggestion (together with previous review comments), some of potential sources discussed in previous manuscript such as river discharge and diffusion of Mn from anoxic continental margin sediments were not appropriate potential source in the South Pacific Gyre. We therefore deleted the corresponding part in this sentence (line 194). Also, we added a statement that hydrogenetic formation of Mn-microparticles primarily occurred in oxic bottom water in the South Pacific Gyre by using manganese derived from eolian transportation in the South Pacific Ocean (lines 167-169).

Ls. 206 ff.: Here you „suddenly“ speak of anoxic sediments. I thought you only investigated oxic sediments?! Moreover, be precise when speaking of anoxic sediments. In most settings, anoxic sediments are found at greater depth while the upper surface of the sediments is characterized by an oxic zone. This zone prevents the diffusive escape of Mn²⁺ from the sediment into the bottom water. Thus, the argumentation in this sentence is imprecise.

Thank you for your valuable comments; discussion on diffusion of dissolved Mn²⁺ from anoxic sediment into the bottom water was imprecise. We deleted the corresponding discussion (line 196).

Concerning the anoxic sediments, we analyzed and compared marine sediment microstructures in various redox condition including oxic and anoxic sediments to understand global distribution of Mn-microparticles discovered in this study. To make this point clear, we rephrased the sentences (lines 196-199).

Ls. 222/223: What precisely do you mean with „the manganese reductive state“ and the „manganese oxidative state“ in this context? The wording is a bit odd. Isn't it rather „oxidized“ or „reduced“?!

Thank you. We agree “oxidized” or “reduced” are correct wordings here. We rephrased the wordings accordingly (lines 201, 211, 217, and 220).

L. 238: „within the sediment“

Corrected (line 226).

Ls. 247 ff.: I would suggest to delete these two sentences because you have no evidence for this assumption. I am sure that measurements of the amount of particulate Mn present over water depth exists in several ocean areas. Are your assumptions in line with such measurements?

We deleted these sentences accordingly (line 233).

L. 265: I do not agree that you have really discussed manganese „dynamics over geologic time“. Therefore, I would suggest to delete this last part of the sentence.

We deleted the last part of the sentence accordingly (line 249).

//